# A dataset of microphysical cloud parameters, retrieved from Emission-FTIR spectra measured in Arctic summer 2017

Philipp Richter[1], Mathias Palm[1], Christine Weinzierl[1], Hannes Griesche[2], Penny M. Rowe[3], and Justus Notholt[1]

[1]University of Bremen, Institute of Environmental Physics, Otto-Hahn-Allee 1, 28359 Bremen, Germany
[2]Leibniz Institute for Tropospheric Research (TROPOS), Permoserstr. 15, 04318 Leipzig, Germany
[3]NorthWest Research Associates, Redmond, WA, USA

**Correspondence:** Philipp Richter (phi.richter@uni-bremen.de)

**Abstract.** A dataset of microphysical cloud parameters from optically thin clouds, retrieved from infrared spectral radiances measured in summer 2017 in the Arctic, is presented. Measurements were performed using a mobile Fourier-transform infrared (FTIR) spectrometer which was carried by the RV *Polarstern*. The dataset contains retrieved optical depths and effective radii of ice and water, from which the liquid water path and ice water path are calculated. The water paths and the effective radii retrieved from the FTIR measurements are compared with derived quantities from a combined cloud radar, lidar and microwave radiometer measurement synergy retrieval, called Cloudnet. The purpose of this comparison is to benchmark the infrared retrieval data against the established Cloudnet retrieval. For the liquid water path, the data correlate, showing a mean bias of $2.48\,\mathrm{g\cdot m^{-2}}$ and a root-mean-square error of $10.43\,\mathrm{g\cdot m^{-2}}$. It follows that the infrared retrieval is able to determine the liquid water path. Although liquid water path retrievals from in the Cloudnet retrieval data come with an uncertainty of at least $20\,\mathrm{g\cdot m^{-2}}$, a root-mean-square error of $9.48\,\mathrm{g\cdot m^{-2}}$ for clouds with a liquid water path of at most $20\,\mathrm{g\cdot m^{-2}}$ is found. This indicates that the liquid water paths especially of thin clouds of the Cloudnet retrieval can be determined with higher accuracy than expected. Apart from this, the dataset of microphysical cloud properties presented here allow researchers to perform calculations of the cloud radiative effects, when the Cloudnet data from the campaign are not available, which was the case from the 22nd July 2017 until the 19th August 2017. The dataset is published at Pangaea (Richter et al., 2021).

## 1 Introduction

Clouds play an important role in the radiation budget of the earth. In the visible regime, clouds mainly reflect and prevent solar radiation from reaching earth's surface, whereas in the thermal regime clouds prevent surface radiation from escaping to space and re-emit it back to earth, where it warms the surface. In the Arctic, about $80\%$ of the liquid water containing clouds have a liquid water path (LWP) below $100\,\mathrm{g\cdot m^{-2}}$ (Shupe and Intrieri, 2004), therefore observation of clouds bearing low amounts of liquid water is crucial to understand the effect of clouds on atmospheric radiation in the Arctic. The change of the

broadband surface longwave radiative flux is largest up to a visible optical depth between 6 to 10 corresponding to a LWP of approximately $40 \, \mathrm{g \cdot m^{-2}}$, depending on the effective droplet radius (Turner et al., 2007).

The observed warming in the Arctic is much greater than the warming of the rest of the Earth (Wendisch et al., 2019). This
phenomenon is called Arctic Amplification. A large number of processes are known to influence the Arctic amplification, but the quantification of each process and its importance is difficult. The project Arctic Amplification: Climate Relevant Atmospheric and Surface Processes and Feedback Mechanisms $(\mathcal{AC})^3$ (Wendisch et al., 2019) aims to close this gap of knowledge by performing various campaigns, model studies and enduring measurements in the Arctic. The measurement campaign and the data presented in this paper are part of $(\mathcal{AC})^3$.

Usually microwave radiometers (MWR) are used for ground-based observations of liquid water clouds. MWR can detect liquid water paths above $100 \, \mathrm{g \cdot m^{-2}}$, also they have the ability to operate continiously 24 hours a day, but LWP retrievals from MWR measurements suffer a high uncertainty in the LWP of at least $15 \, \mathrm{g \cdot m^{-2}}$ (Löhnert and Crewell, 2003). For more accurate observations of optically thin clouds, Fourier Transform Infrared (FTIR) spectrometers can be used. Calibrated FTIR spectrometer are used for the observation of trace gases in absence of the sun or the moon as light source, done for example by
Becker et al. (1999) and Becker and Notholt (2000), as well as for the observation of optically thin clouds, performed within the scope of the network *Atmospheric Radiation Measurement (ARM)* using Atmospheric Emitted Radiance Interferometer (AERI) (Knuteson et al. (2004a) and Knuteson et al. (2004b)). Although the sensitivity of the FTIR retrieval decreases from approximately $50 \, \mathrm{g \cdot m^{-2}}$ (Turner et al., 2007), they can be used to supplement existing cloud observation techniques. In addition, an FTIR spectrometer can be used to determine the effective radii of the cloud droplets and the phase of a cloud. An
emission FTIR spectrometer has been set up on the German research vessel *Polarstern* to perform measurements in summer 2017 in the Arctic around Svalbard.

Lacking freely available physical retrieval algorithms at the time of the measurement campaign, we decided to retrieve microphysical cloud parameters from spectral radiances using the retrieval algorithm Total Cloud Water retrieval (TCWret). TCWret uses the radiative transfer model LBLDIS (Turner, 2005), which includes LBLRTM (Clough et al., 2005) and DIS-
ORT (Stamnes et al., 1988). TCWret works on the spectral radiances from $558.5 \, \mathrm{cm^{-1}}$ to $1163.4 \, \mathrm{cm^{-1}}$, which are taken from (Turner, 2005) and adapted to the present instrumental setup. TCWret uses spectral windows where low absorption of gases occur and therefore the atmosphere is transparent for emissions from clouds. It uses an optimal estimation approach (Rodgers, 2000) and retrieves the liquid water optical depth $\tau_{liq}$, the ice water optical depth $\tau_{ice}$ and their respective effective radii $r_{liq}$ and $r_{ice}$. From this, the LWP and Ice Water Path (IWP) are calculated. The principle of this retrieval technique has been proven
already for mixed-phase clouds by the Mixed-phase cloud property retrieval algorithm (MIXCRA) by Turner (2005) and by the CLoud and Atmospheric Radiation Retrieval Algorithm (CLARRA) by Rowe et al. (2019) and for single-phase liquid clouds using the thermal infrared spectral range (extended line-by-line atmospheric transmittance and radiance algorithm (XTRA) by Rathke and Fischer (2000)).

Section 2 describes the measurement area and gives an overview of the measurement setup and procedure. In section 3, the an-
cillary data from radiosondes and ceilometer are introduced. Section 4 gives a brief description of the infrared retrieval TCWret

**Table 1.** Number of radiance measurements per cruise leg. Only measurements for which there is a successful retrieval are considered.

| Crusie leg | Days with measurement | Measurements |
|:---:|:---:|:---:|
| PS106.1 | 9 | 1746 |
| PS106.2 | 17 | 1915 |
| PS107 | 15 | 1903 |

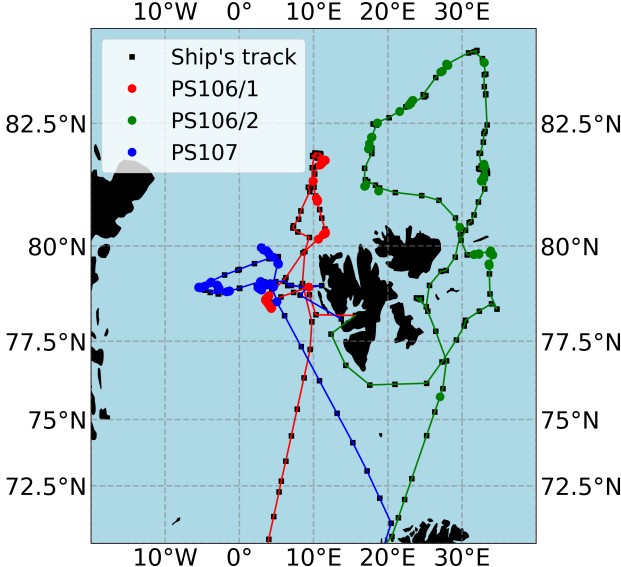

**Figure 1.** Map of the measurement area. Red markers indicate measurements during PS106.1 (24th May 2017 until 21st June 2017), green markers indicate measurements during PS106.2 (23rd June 2017 until 19th July 2017). Blue markers indicate measurements during PS107 (22nd July 2017 until 19th August 2017). The black line shows the ship's track.

and shows the error estimation for this measurement campaign. Section 5 presents the results of the measurement campaign. After the description of data and code availability, a summary and conclusion are provided.

## 2 Observations

### 2.1 Area of Measurements

Measurements were performed around Svalbard from the 24th May 2017 until the 19th August 2017 within the scope of the cruise legs PS106.1 (PASCAL), PS106.2 (SiPCA) and PS107 (FRAM), performed by the RV *Polarstern*. PS106.1 and PS106.2 are collectively referred to as PS106. The cloud cover was observed by meteorologists of the German Weather Service, who

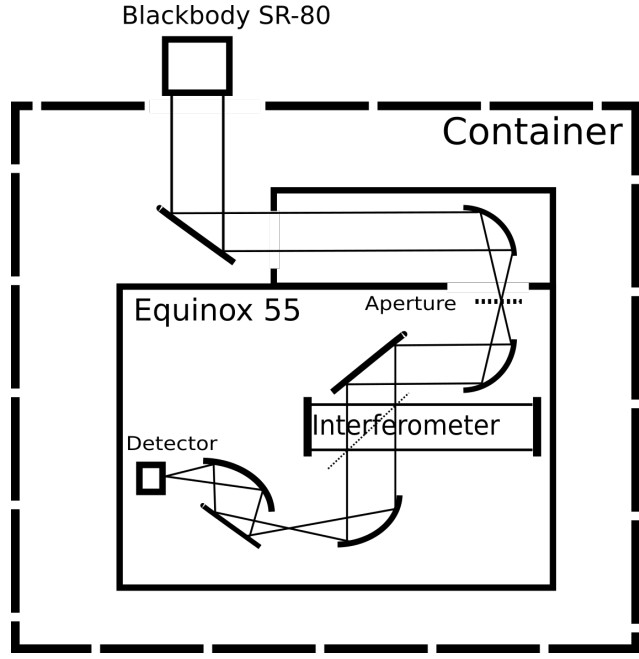

**Figure 2.** Sketch of the *IFS 55 Equinox*. The blackbody SR-80 can be removed, then atmospheric radiation is measured.

**Table 2.** Technical specifications of the FTIR spectrometer IFS 55 Equinox.

| | |
|---|---|
| Beamsplitter | Potassium bromide (KBr) |
| Detector | Mercury-Cadmium-Tellurium (HgCdT) |
| Temperature of Detector | Cooled with liquid nitrogen (77 K) |
| Optical path difference | 3 cm |
| Spectral resolution | $0.3\,\mathrm{cm}^{-1}$ |
| Diameter of entrance arperture | 3.5 cm |

reported a cloud coverage of 7 or 8 oktas in approximately 75% of the time. For further descriptions refer to Macke and Flores (2018) and Schewe (2018). Figure (1) shows the positions of the measurement sites and the ship.

**2.2 Measurement setup**

Measurements of the atmospheric radiances are performed with a mobile FTIR spectrometer (IFS 55 Equinox by Bruker-Optics GmbH) in emission mode (measures atmospheric radiaton without external light source), which will be from now on referred to as EM-FTIR. The instrument was located in an air-conditioned and insulated container on the A-Deck of RV *Polarstern*.

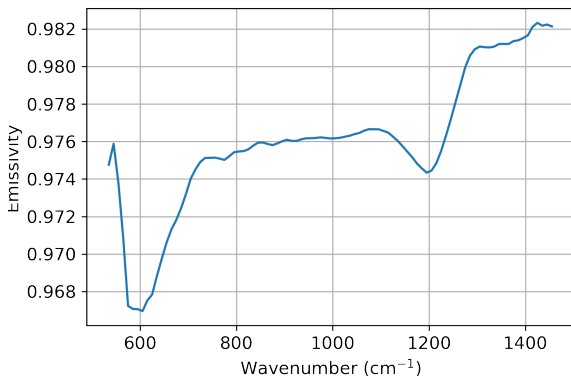

**Figure 3.** Smoothed spectral emissivity of the blackbody radiator.

The roof of the container is two openings. Below one opening the EM-FTIR was located. Both openings can be closed in case of precipitation. The interferometer inside the FTIR spectrometer has a movable mirror giving a maximum optical path difference of $3\,\mathrm{cm}$, which results in a maximum spectral resolution of $\Delta\bar{\nu} = 0.3\,\mathrm{cm}^{-1}$. To prevent damage on the hygroscopic substance of the beamsplitter (Potassium bromoide), the spectrometer is permanently purged with dry air. Further specifications are described in table (2). A blackbody (SR-80 by CI Systems) is placed manually on the EM-FTIR opening at regular intervals to perform a radiometric calibration.

## 2.3   Radiometric calibration and emissivity of the blackbody radiation

To obtain the spectral radiance $L_{atm}$, a radiometric calibration of the EM-FTIR is necessary. To do so, the blackbody radiator SR-80 is used. Its temperature can be set from $-10\,^\circ\mathrm{C}$ to $125\,^\circ\mathrm{C}$. The homogenity of the radiator surface is better than $\pm 0.05\,\mathrm{K}$. The emissivity of the coating is shown in figure (3). The mean emissivity of the blackbody radiator is $\varepsilon = 0.976$. An emissivity below 1 means, that the radiaton of the blackbody is a mixture of the Planck radiation at $T_{BB}$ and the temperature of the container which is assumed to be Planck radiation at $T_{lab}$, The radiation by the EM-FTIR is the sum of the radiation of the radiator plus a term which takes into account the temperature of the environment

$$B = \varepsilon B(T_{BB}) + (1-\varepsilon)B(T_{lab}) \tag{1}$$

with the temperature of the blackbody $T_{BB}$ and the temperature of the laboratory $T_{lab}$, weighted by the blackbody emissivity $\varepsilon$ (Revercomb et al., 1988).

The radiometric calibration of the spectrometer is performed using

$$L_{atm} = \varepsilon B_{\bar{\nu}}(T_{amb}) + \varepsilon \frac{B_{\bar{\nu}}(T_{hot}) - B_{\bar{\nu}}(T_{amb})}{\mathcal{F}(I_{hot} - I_{amb})} \cdot \mathcal{F}(I_{atm} - I_{amb}) + (1-\varepsilon)B_{\bar{\nu}}(T_{lab}) \tag{2}$$

$B(T_{amb,hot,lab})$ are the Planck function at high temperature ($T_{hot}$, setting to $100\,^\circ\mathrm{C}$), surface air temperature ($T_{amb}$) and at the temperature of the laboratory ($T_{lab}$). $I_{hot,amb,atm}$ are the interferograms of the hot blackbody, blackbody at ambient temper-

ature and the atmospheric measurement. $\mathcal{F}$ is the operator for the Fourier transform. In contrast to the procedure described in Revercomb et al. (1988), here the difference of the interferograms is calculated before applying the Fourier transform.

The following cycle is applied for the radiometric calibration: blackbody at $T_{hot}$, atmospheric radiation, blackbody at $T_{amb}$, atmospheric radiation, blackbody at $T_{hot}$ and so on. Each measurement cycle of the blackbodies took about 10 minutes to get one blackbody interferogram $I_{hot}$ or $I_{amb}$. The duration of the atmospheric measurements was approximately 15 minutes. The measurements time and schedule was chosen based on the time it took the blackbody to reach the desired temperature.

## 2.4   OCEANET measurements and Cloudnet synergistic retrieval

Retrievals of microphysical cloud parameters are compared with results of the synergistic retrieval Cloudnet. The OCEANET-Atmosphere observatory from the Leibniz Institute for Tropospheric Research (TROPOS) in Leipzig (Germany) performed continuous measurements during PS106.1 and PS106.2 (Griesche et al., 2020f). Its container houses a multi-wavelength Raman polarization lidar Polly-XT and a microwave radiometer *Humidity and Temperature Profiler (HATPRO)* which was comple-mented during PS106 by a vertically-pointing motion-stabilized 35-GHz cloud radar Mira-35. The OCEANET measurements provide profiles of aerosol and cloud properties and column-integrated liquid water and water vapor content. To retrieve prod-ucts like liquid and ice water content the instrument synergistic approach Cloudnet (Illingworth et al., 2007) was applied to these observations. The retrieved Cloudnet dataset during PS106 has been made available via Pangaea (see table 7). As at-mospheric input, radiosondes launched from the *RV Polarstern* were used. If no radiosonde is available, radiosondes from Ny-Ålesund (if the ship was neas Svalbard) or model data from the Global Data Assimilation System model (GDAS1) were used. A short summary of the Cloudnet retrieval is given in Appendix A. For a detailed description please refer to Griesche et al. (2020f) and the publications cited there.

## 3   Atmospheric profiles and cloud height informations

Auxillary data obained in the ship cruise itself were used to construct the atmospheric setup used in the retrieval. This includes temperature and humidity profiles as well as cloud ceiling measurements.

### 3.1   Cloud ceiling

Information about the cloud ceiling was obtained using a Vaisala Ceilometer CL51 operated by the German Weather Service. The maximum cloud detection altitude is $13\,\mathrm{km}$ with a vertical resolution of $10\,\mathrm{m}$. The uncertainty of the retrieved ceiling is $\pm 1,\%$, but at least $\pm 5\,\mathrm{m}$. Temporal resolution of the results is $60\,\mathrm{s}$. Although only data of the cloud base height is given, it was decided to use these data instead of the Cloudnet height profile, because the ceilometer data was available during the entire cruise, whereas the Cloudnet measurements were only available for the PS106. Without changing the input data, a consistent dataset for the retrieval should be created. However, there is a mean bias between the cloud base height stated by Cloudnet and the ceilometer of $-639\,\mathrm{m}$ (median bias of $-47\,\mathrm{m}$), which means on average the Cloudnet cloud base height is larger than the

ceiling given by the ceilometer, and a root-mean-square error of $1870\,\mathrm{m}$. Data of the ceilometer are available at Schmithüsen
(2017a), Schmithüsen (2017b) and Schmithüsen (2017c).

## 3.2 Radiosounding

Radiosondes were launched four times per day (00 UTC, 06 UTC, 12 UTC, 18 UTC) during the PS106 and twice per day
(06 UTC and 12 UTC) during the PS107 (Schmithüsen (2017d), Schmithüsen (2017e) and Schmithüsen (2017f)). Data were
measured using a RS92 radiosonde by Vaisala. Data of air pressure, temperature, relative humidity, wind speed and wind
direction were recorded. Accuracies are $0.5\,\mathrm{K}$ for temperature measurements, $5\,\%$ for relative humidity and $1\,\mathrm{hPa}$ for air
pressure. Only atmospheric pressure, temperature and humidity were used here. Atmospheric profiles between two radiosonde
launches are acquired by linear interpolation. If a radiosonde stopped measurements before reaching $30\,\mathrm{km}$, data were extended
using the ERA5 reanalysis (Hersbach et al., 2018).

## 4 Total Cloud Water retrieval (TCWret)

**T**otal **C**loud **W**ater **ret**rieval (TCWret) is a retrieval algorithm for microphysical cloud parameters from FTIR spectra. It is
inspired by MIXCRA (Turner, 2005) and XTRA (Rathke and Fischer, 2000) and uses an optimal estimation approach (Rodgers,
2000) to invert the measured spectral radiances for retrieving microphysical cloud parameters. For a complete description of
the retrieval, please refer to appendix B.

### 4.1 Radiative Transfer Models

Two radiative transfer models are used in TCWret: the Line-By-Line Radiative Transfer Model (LBLRTM) (Clough et al.,
2005) and the DIScrete Ordinate Radiative Transfer model (DISORT) (Stamnes et al., 1988). DISORT is called by LBLDIS
(Turner, 2005) to calculate spectral radiances.
LBLRTM calculates the optical depth for gaseous absorbers and the water vapour continuum. The profiles of $H_2O$, $CO_2$, $O_3$,
$CO$, $CH_4$ and $N_2O$ either can be set by the user, or a predefined atmosphere is used. A subarctic summer atmosphere, imple-
mented in LBLRTM, has been used for all gases except $H_2O$, which has been read from radiosonde measurements.
DISORT calculates the monochromatic radiative transfer through a vertically inhomogeneous plane-parallel medium including
scattering, absorption and emission. It provides the spectral radiances using single-scatter parameters.
Several databases are included in LBLDIS (Turner, 2014). These databases contain extinction cross sections, absorption cross
sections, scattering cross sections, single-scattering albedo, asymmetry factor and phase functions for different wavenumbers
and effective radii. Refractive indices for liquid water droplets and ice crystals are taken from Downing and Williams (1975)
and Warren (1984) respectively. Temperature dependent refractive indices for liquid water are from Zasetsky et al. (2005).
However, it is important to note that they have large uncertainties from $1000\,\mathrm{cm^{-1}}$ to $1300\,\mathrm{cm^{-1}}$ (Rowe et al., 2013). Scatter-
ing properties for more complex ice particle shapes like aggregates, bullet rosettes, droxtals, hollow columns, solid columns,
plates and spheroids were calculated by Yang et al. (2001) using a combination of Finite Difference Time Domain (FDTD),

geometric optics and Mie theory.

For all liquid droplets and ice crystals, the droplet size distributions follow a gamma size distribution. The gamma size distributions were chosen in a way, that they fit to the data during the First International Satellite Cloud Climatology Project (ISCCP) Regional Experiment (FIRE) Arctic Cloud Experiment (ACE). For further details, please refer to Turner et al. (2003).

## 4.2   Products of TCWret

Direct retrieval products are $\tau_{liq}$, $\tau_{ice}$, $r_{liq}$ and $r_{ice}$. From these parameters the water paths are calculated:

$$LWP = \frac{2}{3} \cdot r_{liq} \cdot \tau_{liq} \cdot \varrho_{liq} \tag{3}$$

$$IWP = \frac{N \cdot V_0(r_{ice}) \cdot \tau_{ice}}{\alpha_{ice}} \cdot \varrho_{ice} \tag{4}$$

with the volumetric mass densities of liquid water $\varrho_{liq} = 1000\,\mathrm{kg} \cdot \mathrm{m}^{-3}$, ice water $\varrho_{ice} = 917\,\mathrm{kg} \cdot \mathrm{m}^{-3}$, the particle number density $N$ and the extinction coefficient $\alpha_{ice} = \beta_{ice} \cdot N$. The total volume of an ice crystal $V_0(r_{ice})$ and the extinction cross

section of an ice droplet $\beta_{ice}$, both integrated over the gamma size distribution are read from the databases of single-scattering parameters. The formula for the liquid water path works for spherical droplet only, while the formula for the ice water path is valid for ice crystals of any shape (Turner, 2005). The covariance matrix $\mathbf{S}_r$ of the optimal estimation procedure is used to determine the errors.

## 4.3   Covariance matrix and averaging kernels

Retrieval errors are calculated from the variance-covariance matrix $\mathbf{S}_r$ of the retrieval. It is calculated by

$$\mathbf{S}_r = \mathbf{T}_r \mathbf{S}_y \mathbf{T}_r^T \tag{5}$$

The index $r$ denotes quantities of the final iteration. $\mathbf{T}$ is a transfer matrix and $\mathbf{S}_y$ is the variance-covariance matrix of the measurement. The retrieval uses a Levenberg-Marquardt algorithm, therefore the variance-covariance matrix and the transfer matrix $\mathbf{T}$ are calculated iteratively, as described by Ceccherini and Ridolfi (2010). Another important quantity to characterize

the retrieval quality is the Averaging Kernel Matrix $\mathbf{A}$. The averaging kernel matrix contains the derivatives of the retrieved quantities with respect to the true state vector

$$\mathbf{A} = \frac{\partial \boldsymbol{x_r}}{\partial \boldsymbol{x_t}} \tag{6}$$

where $\boldsymbol{x_r}$ means the retrieved parameters and $\boldsymbol{x_t}$ are the unknown true parameters. In TCWret, the averaging kernel matrix is a $4 \times 4$-matrix. The top two rows belong to $\tau_{liq}$ and $\tau_{ice}$, the bottom two rows belong to $r_{liq}$ and $r_{ice}$. On the diagonal elements

one finds the derivatives of each element in the retrieved state vector with respect to its corresponding element in the true state

vector. Off-diagonal elements give the degree of correlation between the entries of the state vector

$$\mathbf{A} = \begin{pmatrix} A_{\tau_{liq}} & A_{\tau_{liq},\tau_{ice}} & A_{\tau_{liq},r_{liq}} & A_{\tau_{liq},r_{ice}} \\ A_{\tau_{ice},\tau_{liq}} & A_{\tau_{ice}} & A_{\tau_{ice},r_{liq}} & A_{\tau_{ice},r_{ice}} \\ A_{r_{liq},\tau_{liq}} & A_{r_{liq},\tau_{ice}} & A_{r_{liq}} & A_{r_{liq},r_{ice}} \\ A_{r_{ice},\tau_{liq}} & A_{r_{ice},\tau_{ice}} & A_{r_{liq},r_{ice}} & A_{r_{ice}} \end{pmatrix} \tag{7}$$

Here $A_{v,w}$ stands for the mutual dependence of the parameters $v$ and $w$, where $v$ is the parameter in $\boldsymbol{x_r}$ and $w$ is the parameter in $\boldsymbol{x_t}$. The trace of the averaging kernel matrix gives the degrees of freedom of the signal, which can be interpreted as the number of individually retrievable parameters from the measurement (Rodgers, 2000). The averaging kernel matrix sets the retrieval and the a priori into context:

$$\boldsymbol{x_r} = \boldsymbol{x_a} + \mathbf{A}(\boldsymbol{x_t} - \boldsymbol{x_a}) \tag{8}$$

From this relationship it can be seen that in the optimal case the Averaging Kernel Matrix is the unit matrix. Smaller entries mean a stronger influence by the a priori. Averaging kernels in TCWret are calculated via

$$\mathbf{A} = \mathbf{T}_r \mathbf{K}_r \tag{9}$$

The matrix $\mathbf{K}_r$ is the jacobian matrix of the retrieved parameters (Ceccherini and Ridolfi, 2010). Uncertainties of LWP and IWP are calculated from error propagation:

$$\sigma_Y = \pm \sqrt{\sum_i \left( \frac{\partial Y}{\partial m_i} \sigma_{m_i} \right)^2} \tag{10}$$

where $Y$ is either LWP or IWP, $\frac{\partial Y}{\partial m}$ is the partial derivative of $Y$ with respect to an atmospheric parameter $m = \{\tau_{liq}, \tau_{ice}, r_{liq}, r_{ice}\}$ and $\sigma_{m_i}$ is the variance of the $i$-th parameter $m_i$, as stated in $\mathbf{S_r}$.

### 4.4 Performance of TCWret applying to simulated data

In addition to the uncertainties indicated by the optimal estimation procedure, TCWret was applied to simulated data (Cox et al., 2016). The description of the testcases and the evaluation can be found in the appendix C. Results are shown in table (3). When applied to the simulated data, it could be shown that TCWret can determine all variables entered in the table. Results calculated by TCWret are comparable to the true cloud parameters from the simulated data.

### 4.5 Erorrs of atmospheric profile and calibration

Besides the uncertainties from the optimal estimation algorithm, uncertainties from atmospheric profile data and the calibration cycle increase the total uncertainty of the data.

**Table 3.** Results of the testcase retrievals. $|r|$ is the correlation coefficient of each quantity. Mean bias is the mean difference between retrieval and the true size of the parameter. RMSE is the root-mean-square of the difference between retrieval and true parameter. For $\tau_{liq,ice}$ and $r_{liq,ice}$, ERR (OE) is the standard deviation calculated from the posterior coviarance matrix of the optimal estimation, stated in equation (5). For the other quantities, ERR(OE) is calculated by error propagation. *Maximum of quantity in testcases specifies* the maximum value that can be used for this quantity in the test cases. A total number of 253 testcases are included in these calculations.

| Quantity | $\|r\|$ | Mean Bias | RMSE | ERR (OE) | Maximum of quantity in testcases |
|---|---|---|---|---|---|
| $\tau_{liq}$ (1) | 0.86 | −0.1 | 0.5 | 0.3 | 5.45 |
| $\tau_{ice}$ (1) | 0.78 | 0.2 | 0.6 | 0.3 | 4.45 |
| $\tau_{cw} = \tau_{liq} + \tau_{ice}$ (1) | 0.99 | 0.1 | 0.2 | 0.7 | 5.94 |
| $f_{ice}$ (1) | 0.70 | 0.08 | 0.3 | 0.6 | 1.0 |
| $r_{liq}$ (µm) | 0.59 | −2.4 | 4.1 | 2.9 | 22.00 |
| $r_{ice}$ (µm) | 0.65 | 3.0 | 10.0 | 2.4 | 70.00 |
| LWP (g·m$^{-2}$) | 0.68 | −1.9 | 6.3 | 2.3 | 46.90 |
| IWP (g·m$^{-2}$) | 0.82 | 1.9 | 10.0 | 5.1 | 107.39 |

**Table 4.** Mean partial derivatives, used for estimating the parameter errors $\Delta par$.

| Quantity $m$ | $\frac{\partial m}{\partial T}$ | $\frac{\partial m}{\partial q}$ | $\frac{\partial m}{\partial L}$ |
|---|---|---|---|
| $\tau_{liq}$ | 0.03 | 0.02 | −0.01 |
| $\tau_{ice}$ | 0.12 | −0.01 | −0.02 |
| $r_{liq}$ | −0.58 | 0.14 | −1.97 |
| $r_{ice}$ | 1.38 | 0.62 | −7.01 |
| LWP | −0.27 | 0.15 | −0.47 |
| IWP | 2.43 | 0.14 | −1.41 |

### 4.5.1 Partial derivatives for non-retrieved quantities

To estimate the uncertainty which comes from the cloud temperature, humidity profile and spectral calibration, the testcases from Cox et al. (2016) have been adjusted to incorporate uncertainties in cloud temperature, humidity and radiance. Three datasets are creating, each of them with one of the following adjustments:

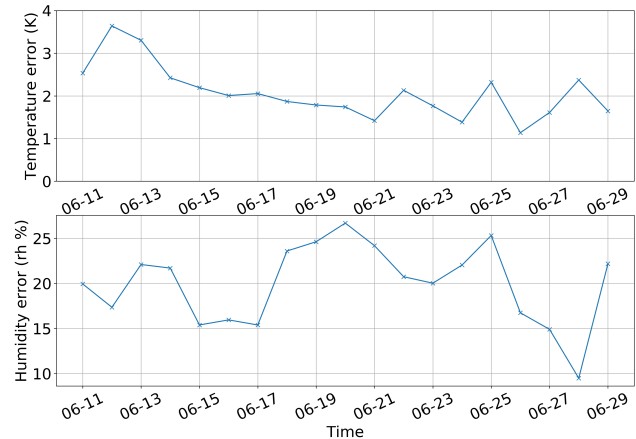

**Figure 4.** Total error as sum of device error and interpolation error.

    – Increase cloud temperature by $1\,\text{K}$

    – Increase atmospheric humidity by $10\,\%$

– Increase radiance by $2\,\text{mW} \cdot (\text{cm}^{-1} \cdot \text{m}^2 \cdot \text{sr})^{-1}$

With these datasets the partial derivatives are calculated, which are necessary to determine the errors due to cloud temperature, humidity and spectral calibration and propagate them into the retrieved cloud parameters by application of

$$\Delta m = \pm\sqrt{\left(\frac{\partial m}{\partial T}\Delta T\right)^2 + \left(\frac{\partial m}{\partial q}\Delta q\right)^2 + \left(\frac{\partial m}{\partial L}\Delta L\right)^2} \tag{11}$$

with the cloud temperature $T$, the relative humidity $q$, the radiance $L$ and their errors $\Delta T$, $\Delta q$ and $\Delta L$. To separate the influence
of the parameter errors from the retrieval performance, the results of these three datasets are compared to the retrieval results mentioned in section 4.3 instead of the true cloud parameters. Mean partial derivatives are then calculated as follows:

1. Retrieve the cloud parameters for each dataset

2. Calculate the difference between the cloud parameters of the adjusted dataset and the undisturbed dataset (which has been already used in section 4.3)

3. Calculate the difference quotients, which will act as partial derivates in equation (11)

The partial derivatives are shown in table (4).

### 4.5.2 Temperature and humidity

Device errors of the radiosonde are $\Delta T = 0.5\,\text{K}$ and $\Delta q = 5\,\%$. Additionally, the error introduced with the linear interpolation of the temperature and relative humidity is estimated by comparing the interpolated profiles to atmospheric profiles from

220 ERA5. The interpolation error follows from the comparison between the linear interpolation between two radiosonde measurements and the ERA5 atmosphere at the position of the measurements. We query the ERA5 atmosphere for each hour. Then we calculate the atmospheric profiles from the radiosondes once per hour by linear interpolation. From this we calculate the difference, average over one day and calculate the standard deviation. Figure (4) gives the total error as device error and interpolation error, as an example for the period from 11th June 2017 to 30th June 2017.

### 4.5.3 Calibration error

The accuracy of the blackbody temperature and emissivity are $\Delta T_{BB} = \pm 0.05\,\text{K}$ and $\Delta \epsilon = \pm 0.02$. The propagation of these errors into the radiance is

$$\Delta L = \sqrt{\left(\frac{\partial L_{atm}}{\partial \epsilon} \cdot 0.02\right)^2 + \left(\frac{\partial L_{atm}}{\partial T_{BB}} \cdot 0.05\,\text{K}\right)^2} \tag{12}$$

To estimate $\frac{\partial L_{atm}}{\partial \epsilon}$, a spectrum is calibrated with an emissivity of $\epsilon'$ and $\epsilon' + h$. The partial derivative is calculated by $\frac{\partial L_{atm}}{\partial \epsilon} = \frac{L(\epsilon' + h) - L(\epsilon')}{h}$ with $L(\epsilon')$, the radiance under the emissivity $\epsilon'$ and $h$ as step size for the numerical calculation of the partial derivative. From $\epsilon' = 0.975$ and $h = 0.02$ follows $\frac{\partial L_{atm}}{\partial \epsilon} \cdot 0.02 = -0.98\,\text{mW} \cdot (\text{sr} \cdot \text{cm}^{-1} \cdot \text{m}^2)^{-1}$. The second partial derivative $\frac{\partial L_{atm}}{\partial T_{BB}}$ is estimated using equation (2). The emissivity is set to 1. The measured radiance of the hot blackbody is larger than the radiance of the atmosphere ($\mathcal{F}(I_{hot}) > \mathcal{F}(I_{atm})$) and therefore the quotient

$$\frac{\mathcal{F}(I_{atm} - I_{amb})}{\mathcal{F}(I_{hot} - I_{amb})} < 1 \tag{13}$$

From the measurements it follows that $L_{hot}$ is about five times larger than $L_{amb}$, therefore there inequtaion (13) is set $\frac{\mathcal{F}(I_{hat} - I_{amb})}{\mathcal{F}(I_{hot} - I_{amb})} = 0.2$. Equation (2) thus can be written as

$$L_{atm} = B_{\bar{\nu}}(T_{amb}) + 0.2 \cdot B_{\bar{\nu}}(T_{hot}) - B_{\bar{\nu}}(T_{amb}) \tag{14}$$

With $T_{BB} = T_{hot} = 100\,^\circ\text{C}$ and $T_{amb} = 0\,^\circ\text{C}$ is $\frac{\partial L_{atm}}{\partial T_{BB}} \cdot 0.05 = 0.10\,\text{mW} \cdot (\text{sr} \cdot \text{cm}^{-1} \cdot \text{m}^2)^{-1}$ as an average for the spectral
interval between $500\,\text{cm}^{-1}$ and $2000\,\text{cm}^{-1}$. This gives $\Delta L = 0.98\,\text{mW} \cdot (\text{sr} \cdot \text{cm}^{-1} \cdot \text{m}^2)^{-1}$.

### 4.5.4 Resulting parameter error

Finally, from the calculations in this section, the resulting unvertainties are

- $\Delta T = 2.0\,\text{K}$, as sum of the device error ($0.5\,\text{K}$) and the interpolation error ($1.5\,\text{K}$)

- $\Delta q = 17.5\%$, as sum of the device error ($5.0\%$) and the interpolation error ($12.5\%$)

- $\Delta L = 0.98\,\text{mW} \cdot (\text{sr} \cdot \text{cm}^{-1} \cdot \text{m}^2)^{-1}$

Applying these uncertainties to equation (11), the uncertainties for each parameter are $\Delta\tau_{liq} = 0.4$, $\Delta\tau_{ice} = 0.3$, $\Delta r_{liq} = 3.3\,\mu\text{m}$, $\Delta r_{ice} = 13.1\,\mu\text{m}$, $\Delta LWP = 2.8\,\text{g} \cdot \text{m}^{-2}$ and $\Delta IWP = 5.6\,\text{g} \cdot \text{m}^{-2}$. These values will be added to the retrieval errors in the next section.

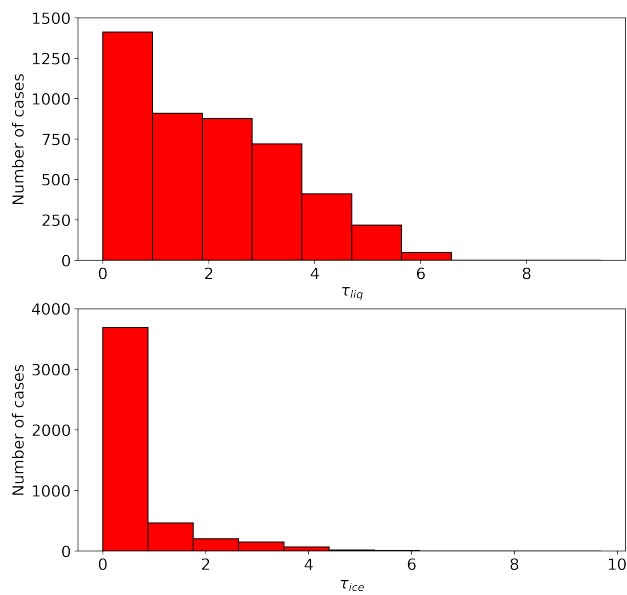

**Figure 5.** Distribution of retrieved optical depths for liquid water (upper panel) and ice water (lower panel). The binwidth is set to the sum of the root-mean-square from table (3) and the errors discussed in section 4.5.

**Table 5.** Key features of the dataset

| Key figure | Size |
|---|---|
| Retrievals performed | 5564 |
| Accepted retrievals | 4590 |
| Mixed-phase clouds ($0.1 < f_{ice} < 0.9$) | 2158 |
| Single-phase liquid ($f_{ice} < 0.1$) | 2899 |
| Single-phase ice ($f_{ice} > 0.9$) | 507 |
| Minimum observed precipitable water vapour (PWV) | 0.67cm |
| Maximum observed precipitable water vapour (PWV) | 1.62cm |

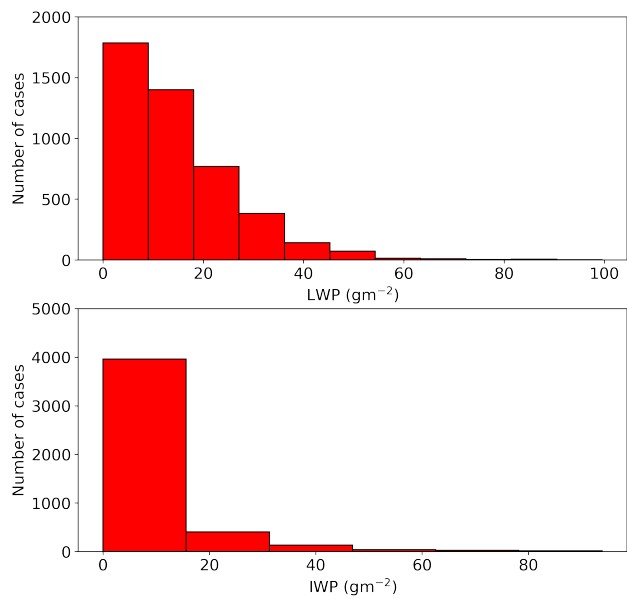

**Figure 6.** Distribution of retrieved LWP (upper panel) and IWP (lower panel). The binwidth is set to the sum of the root-mean-square from table (3) and the errors discussed in section 4.5.

## 5 Results

### 5.1 Cloud parameters from infrared radiance measurements during the PS106 and PS107

During the measurement campaign, most of the observed optical depth is due to liquid water insted of ice crystals. A histogram of all retrieved optical depths are shown in figure (5). In $66.4\%$ of the measurements, ice was observed in the clouds, whereas in $92.4\%$ of the measurements liquid water was present. Mean optical depths are $\tau_{liq} = 2.6$ and $\tau_{ice} = 0.8$. Similar to the optical depth, most of the observed cloud water is liquid water (figure 6). Here the means are $LWP = 17.7\,\mathrm{g}\cdot\mathrm{m}^{-2}$ and $IWP = 9.9\,\mathrm{g}\cdot\mathrm{m}^{-2}$. Interquartile ranges for LWP and IWP are $IQR_{LWP} = 18.9\,\mathrm{g}\cdot\mathrm{m}^{-2}$ and $IQR_{IWP} = 11.5\,\mathrm{g}\cdot\mathrm{m}^{-2}$. Whereas the range of LWP matches the LWP from the testcases, the IWP is near the lower threshold of the retrievable water path.

The distributions of the effective radii are shown in figure (7). For $r_{liq}$ only cases with $f_{ice} < 0.9$ are used and for $r_{ice}$ only cases with $f_{ice} > 0.1$ are used. On average, ice crystals ($r_{ice} = 22.3\,\mathrm{\mu m}$) are larger than liquid droplets ($r_{liq} = 10.9\,\mathrm{\mu m}$). Ice crystals show a wider range of retrieved effective radii than liquid droplets, expressed by an interquartile range of $IQR_{ice} = 17.9\,\mathrm{\mu m}$ compared to $IQR_{liq} = 5.9\,\mathrm{\mu m}$.

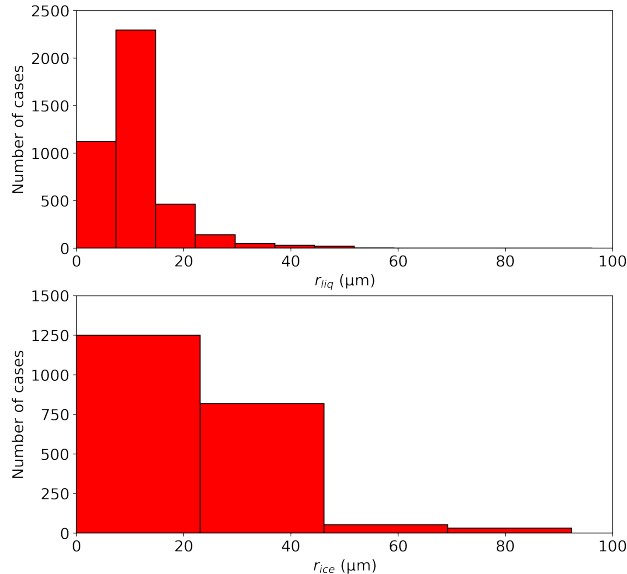

**Figure 7.** Distribution of retrieved effective radii for liquid water droplets (upper panel) and ice crystals (lower panel). The binwidth is set to the sum of the root-mean-square from table (3) and the errors discussed in section 4.5. In each case, only cases are considered in which the phase fractions are above 0.1 (Liquid water fraction for $r_{liq}$ and ice water fraction for $r_{ice}$). This results in 4111 of 4590 cases for $r_{liq}$ (89.6%) and 2153 of 4590 cases for $r_{ice}$ (46.9%).

## 5.2 Averaging Kernels and posterior correlation matrices

For all measurements, the mean of the averaging kernels and degrees of freedom are calculated:

$$
\mathbf{A} = \begin{pmatrix} 0.77 & 0.48 & -0.17 & -0.02 \\ 0.19 & 0.45 & 0.25 & -0.01 \\ -0.04 & 0.14 & 0.74 & 0.05 \\ -0.03 & -0.1 & 0.29 & 0.3 \end{pmatrix} \tag{15}
$$

$$
tr(\mathbf{A}) = 2.25 \tag{16}
$$

This mean averaging kernel matrix contains both single-phase clouds and mixed-phase clouds. Since only two parameters are determined in the single-phase cases, they perturb the mean number of degrees of freedom for all measurements. As seen in the statistics, there are less cases with ice-containing clouds. This lowers the entries on the diagonals for $\tau_{ice}$ and $r_{ice}$ as they

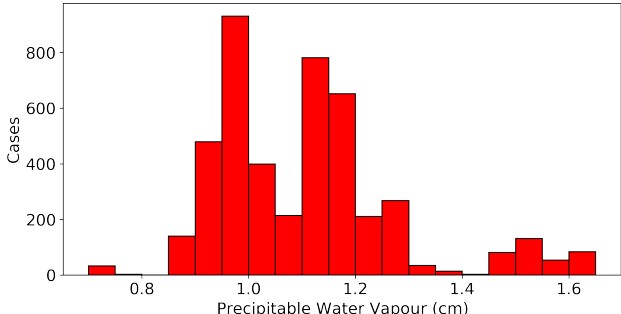

**Figure 8.** Histogram of the precipitable water vapour during the measurements of atmospheric radiances.

are 0 in all-liquid clouds. Therefore, the mean averaging kernel was also calculated for all mixed-phase clouds:

$$\mathbf{A}_{mixed-phase} = \begin{pmatrix} 0.62 & 0.22 & -0.35 & -0.03 \\ 0.32 & 0.7 & 0.47 & -0.04 \\ -0.08 & 0.16 & 0.66 & 0.1 \\ -0.14 & -0.07 & 0.17 & 0.59 \end{pmatrix} \tag{17}$$

$$tr(\mathbf{A}_{mixed-phase}) = 2.57 \tag{18}$$

The number of degrees of freedom in this case is 2.57. The entries for the effective radii are at the same size as those for the optical depth.

The posterior correlation matrix $\mathbf{R}$ gives the correlations of one retrieved parameter to another. For mixed-phase clouds, $\mathbf{R}$ is

$$\mathbf{R}_{mixed-phase} = \begin{pmatrix} 1.00 & 0.50 & -0.07 & -0.40 \\ 0.50 & 1.00 & 0.02 & -0.23 \\ -0.07 & 0.02 & 1.00 & 0.13 \\ -0.41 & -0.23 & 0.13 & 1.00 \end{pmatrix} \tag{19}$$

Largest correlation appear between $\tau_{liq}$ and $\tau_{ice}$ ($|r| = 0.50$), which points to a difficult phase determination. Apart from the correlation of the optical thicknesses, the comparatively high correlation between $r_{ice}$ and $\tau_{liq}$ is striking, which suggests that both parameters cannot be determined completely independently of each other.

**5.3 Precipitable water vapour**

A crucial spectral region for the determination of the cloud phase is the spectral window in the far-infrared between $500\,\mathrm{cm}^{-1}$ and $600\,\mathrm{cm}^{-1}$ (Rathke et al., 2002). This spectral region is sensitive to the concentration of water vapour in the atmosphere. The amount of water vapour is expressed by the precipitable water vapour PWV, which has been calculated from the radiosonde measurements. The far-infrared spectral region becomes nearly opaque to infrared radiation for PWV $> 1\,\mathrm{cm}$ (Cox et al.,

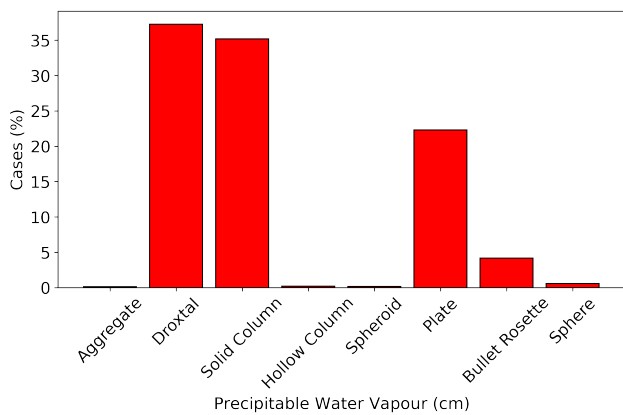

**Figure 9.** Percentage of retrievals for each ice particle shape. Most particles are modelled as droxtals (37%), solid columns (35%), plates (22%) and bullet rosettes (4%)

**Table 6.** Results of the comparison between TCWret and Cloudnet. Mean Bias and Root-Mean-Square error refer to the difference of both datasets.

| Quantity | $\lvert r \rvert$ | Mean Bias | Root-Mean-Square error |
|---|---|---|---|
| LWP | 0.65 | $2.5\,\mathrm{g}\cdot\mathrm{m}^{-2}$ | $10.4\,\mathrm{g}\cdot\mathrm{m}^{-2}$ |
| LWP $< 20\,\mathrm{g}\cdot\mathrm{m}^{-2}$ | 0.52 | $4.0\,\mathrm{g}\cdot\mathrm{m}^{-2}$ | $9.5\,\mathrm{g}\cdot\mathrm{m}^{-2}$ |
| LWP $(PWV < 1\,\mathrm{cm})$ | 0.73 | $1.1\,\mathrm{g}\cdot\mathrm{m}^{-2}$ | $8.3\,\mathrm{g}\cdot\mathrm{m}^{-2}$ |
| LWP $< 20\,\mathrm{g}\cdot\mathrm{m}^{-2}$ $(PWV < 1\,\mathrm{cm})$ | 0.7 | $2.6\,\mathrm{g}\cdot\mathrm{m}^{-2}$ | $5.9,\mathrm{g}\cdot\mathrm{m}^{-2}$ |
| IWP | 0.41 | $1.5\,\mathrm{g}\cdot\mathrm{m}^{-2}$ | $16.8\,\mathrm{g}\cdot\mathrm{m}^{-2}$ |
| $r_{liq}$ | 0.68 | $4.5\,\mathrm{\mu m}$ | $5.3\,\mathrm{\mu m}$ |
| $r_{liq}$ (maximum) | 0.69 | $3.1\,\mathrm{\mu m}$ | $4.2\,\mathrm{\mu m}$ |

2015). During the measurement campaign the PWV was greater than $1\,\mathrm{cm}$ in $62\%$ of the cases. Therefore, the datasets for PWV greater than $1\,\mathrm{cm}$ are not removed from the analysis. Statistics of PWV are shown in figure (8).

### 5.4 Comparison to Cloudnet

To compare result from TCWret and Cloudnet, a combined dataset of TCWret results is created in the following way: Since the shapes of the ice crystals are not known, the retrievals were carried out for all ice crystal shapes. However, this procedure
leads to up to 8 results per measurement, so a selection was made. The aim of the following selection is that all ice crystals with

$r_{ice} < 30\,\mu\text{m}$ are modelled as droxtals, while larger ice crystals are modelled as either plates, bullet rosettes or solid columns. This choice is motivated by Yang et al. (2007). The accepted result then is determined as follows:

1. If $r_{ice}$ for plates, bullet rosettes and solid columns or for droxtals are less than $30\,\mu\text{m}$, the result using ice crystals as droxtals is accepted.

2. If $r_{ice}$ for droxtals are greater than $10\,\mu\text{m}$, the result that uses plates, bullet rosettes or solid columns is accepted. To choose one of the datasets, a random number is drawn which selects plates in $35\%$, bullet rosettes in $15\%$ and solid columns in $50\%$.

3. If none of the conditions apply, the data for which the degrees of freedom of the outcome are highest is accepted.

The first condition ensures that all small ice particles are classified as droxtals, while the second ensures that all larger particles are classified as plates, solid columns or bullet rosettes. Stricter thresholds would more often result in only the last condition applying, which should be avoided as much as possible.

As additional constraint, we only allow results where $r_{liq} < r_{ice}$. This is motivated by the following: The results of $r_{liq}$ and $r_{ice}$ show that $r_{liq}$ is usually smaller than $r_{ice}$. This applies to both TCWret and Cloudnet. Therefore, cases with $r_{liq} > r_{ice}$ are likely cases with a too small $r_{ice}$ and a too large $r_{liq}$. For the comparison between TCWret and Cloudnet, results from both datasets were averaged over a time period of two minutes. This has been done because the underlying measurement systems have different temporal resolutions, also both measurement systems were at different locations on the ship. Cloudnet results do not contain optical depths, but water paths and droplet radii, therefore we will compare LWP and IWP, $r_{liq}$ and $r_{ice}$. Correlation coefficients, mean biases and root-mean-square errors are shown in table (6).

### 5.4.1 Ice Water Path and ice effective radius

Although TCWret can determine $r_{ice}$ from the simulated spectra, no correlation can be found between the TCWret and Cloudnet data. From the error considerations in previous sections it was shown that the RMSE for the simulated spectra is already $10.0\,\mu\text{m}$. Taking into account uncertainties in the atmospheric data and the calibration, an additional uncertainty term of $13.1\,\mu\text{m}$ is obtained, so that $r_{ice}$ is already subject to high uncertainties. According to the posterior correlation matrix, $r_{ice}$ correlates with $\tau_{liq}$, so that there is no completely independent result of $r_{ice}$. A better determination of $r_{ice}$ could be achieved by a better A priori $x_a$, but the problem remains that according to the averaging kernel matrix only $2.57$ degrees of freedom exist in the measurements.

Figure (10) shows the results for the IWP. Although a correlation can be found, there is large spread between the datasets. The difference between TCWret and Cloudnet is $(1.5 \pm 16.8)\,\text{g} \cdot \text{m}^{-2}$. The IWP is calculated according to equation (4) from $\tau_{ice}$ and $r_{ice}$, where $r_{ice}$ influences the IWP of the TCWret data set. Furthermore, the IWP during the measurement campaign is $9.9\,\text{g} \cdot \text{m}^{-2}$, very low and within the RMSE of TCWret when retrieving the simulated spectra. The IWP is therefore at the lower limit of what can be determined with TCWret and is considered as less reliable.

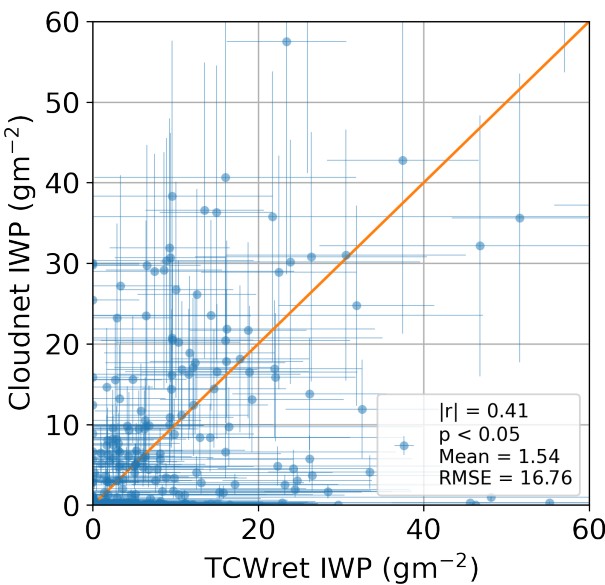

**Figure 10.** Ice water path of TCWret versus IWP from Cloudnet.

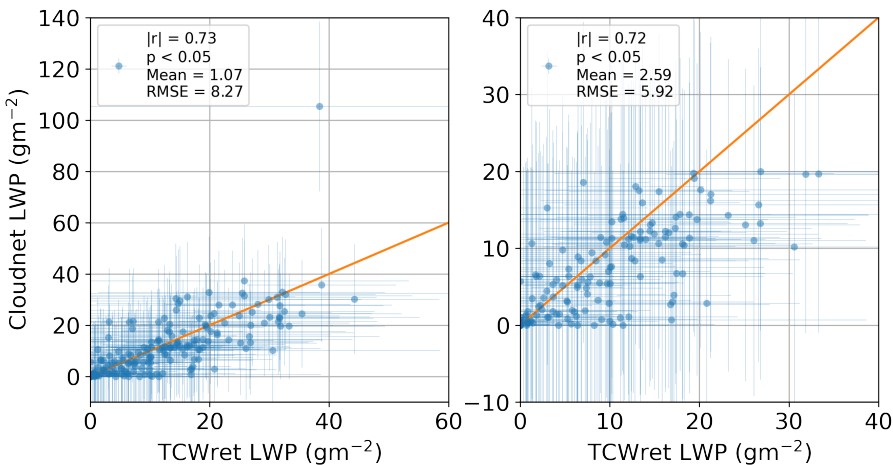

**Figure 11.** Liquid water path of TCWret versus Cloudnet for $PWV < 1\,\mathrm{cm}$. Left scatter plot contains all measurements, whereas the right plot only shows clouds with LWP $< 20\,\mathrm{g\cdot m^{-2}}$.

### 5.4.2 Liquid Water Path and effective droplet radius

Results of liquid water path from TCWret and Cloudnet are correlated. The difference is $2.5\,\mathrm{g\cdot m^{-2}} \pm 10.4\,\mathrm{g\cdot m^{-2}}$ with no restriction to the maximum water path. From this we conclude that the LWP from the TCWret-dataset is reliable. As mentioned earlier, a large PWV interferes with the retrieval, as the water vapour has a larger influence on the microwindows. Therefore, we

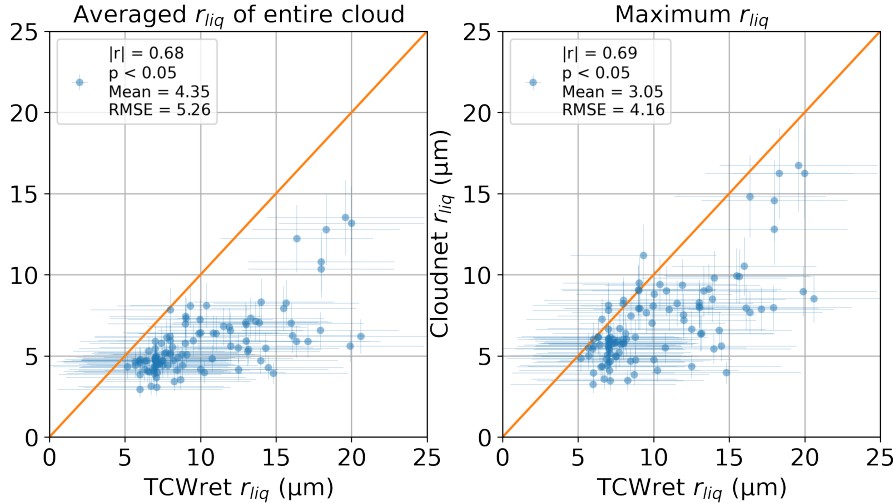

**Figure 12.** $r_{liq}$ of TCWret versus $r_{liq}$ from Cloudnet averaged over the entire cloud (left panel) and maximum value of the cloud from Cloudnet.

additionally remove all cases from the analysis, where the PWV is larger than $1\,\mathrm{cm}$. This reduces the mean bias to $1.1\,\mathrm{g\cdot m^{-2}}$ and the RMSE to $8.3\,\mathrm{g\cdot m^{-2}}$. The results with PWV $< 1\,\mathrm{cm}$ are shown in figure (11, left panel).

Since the LWP of TCWret correlates with that of the Cloudnet product, and since the RMSE of the LWP is far below the uncertainty of the LWP of the Cloudnet product, we reduced the maximum LWP to investigate whether a correlation can
also be observed for clouds with an $LWP < 20\,\mathrm{g\cdot m^{-2}}$. With a real uncertainty of $\pm 20\,\mathrm{g\cdot m^{-2}}$ the correlation is expected to disappear.

Results for very thin clouds and PWV $< 1\,\mathrm{cm}$ are shown in figure (11) (right side). Again, results are correlated. The RMSE for these clouds is $5.9\,\mathrm{g\cdot m^{-2}}$ with a mean bias of $2.6\,\mathrm{g\cdot m^{-2}}$. Without any restrictions on the PWV, there is a RMSE of $9.5\,\mathrm{g\cdot m^{-2}}$ and a mean bias of $4.0\,\mathrm{g\cdot m^{-2}}$. From the comparison with TCWret, it can be concluded that during this measurement campaign,
Cloudnet's results for thin clouds with LWP $< 20\,\mathrm{g\cdot m^{-2}}$ are also reliable despite the stated error of $20.40\,\mathrm{g\cdot m^{-2}}$.

It should be noted that Cloudnet and TCWret use the atmospheric profiles from the radiosonde measurements carried out on the RV Polarstern. Apart from that, however, both the measuring instruments and the retrievals are different. Furthermore, TCWret does not use information from Cloudnet as A Priori. Since TCWret has also shown comparable agreement with the LWP of the simulated spectra in the test cases (mean bias is $-1.6\,\mathrm{g\cdot m^{-2}}$, RMSE is $6.3\,\mathrm{g\cdot m^{-2}}$), it is to be expected that TCWret and thus
also Cloudnet have independently determined the LWP correctly.

Figure (12) shows the results for $r_{liq}$. The left panel shows the results where $r_{liq}$ of Cloudnet is averaged over the entire cloud. The right panel shows the maximum $r_{liq}$ of the cloud in the Cloudnet data. Only results from TCWret are considered if $f_{ice} < 0.9$. As in the LWP, a correlation between the data can be observed. Overall, there is an overestimation of the $r_{liq}$ of TCWret by $4.4\,\mu\mathrm{m}$ on average. If only considering the maximum $r_{liq}$ in Cloudnet, the mean bias decreases to $3.1\,\mu\mathrm{m}$. The

same applies to the RMSE, which decreases from $5.3\,\mu\mathrm{m}$ to $4.2\,\mu\mathrm{m}$. These results indicate that $r_{liq}$ in TCWret does not take into account the entire cloud, which is to be expected since the $r_{liq}$ in Cloudnet is determined using the altitude-resolved radar reflectivity, while TCWret uses the radiance of the clouds measured on the ground. However, the observed correlation allows a correction of $r_{liq}$ in TCWret as a function of $r_{liq}$ itself.

## 6 Data availability

For accessibility of used and shown datasets, see table (7).

## 7 Code availability

The retrieval algorithm TCWret is available at https://doi.org/10.5281/ZENODO.4621127 (Richter, 2021) with external sub-routines at https://doi.org/10.5281/ZENODO.4618142 and https://doi.org/10.5281/ZENODO.4618106. Jupyter-Notebooks to perform the comparisons to Cloudnet are available at https://github.com/RichterIUP/evaluation_tcwret.

## 8 Summary and Conclusion

A dataset of microphysical cloud parameters of optically thin clouds is presented. The measurements were carried out on the ship RV *Polarstern* in summer 2017 in the Arctic Ocean around Svalbard and in the Fram Strait.

Measurements were performed using a mobile FTIR spectrometer, operated in emission mode (EM-FTIR). A calibration of the EM-FTIR was performed with a blackbody radiator, whose temperature was alternately set to $100\,^\circ\mathrm{C}$ and ambient temperature. The spectrometer was operated in an air-conditioned container. Radiances between $500\,\mathrm{cm}^{-1}$ and $2000\,\mathrm{cm}^{-1}$ were recorded. The retrieval of cloud parameters is performed using the Total Cloud Water retrieval (TCWret). TCWret uses the optimal estimation method to invert atmospheric radiances. The radiative transfer model used is LBLDIS, which utilizes optical depths of atmospheric trace gases calculated with LBLRTM and then calculates the spectral radiances using DISORT. Single-scattering parameters for clouds are read from pre-calculated databases. Retrieval products are the optical depths of water and ice and the corresponding effective radii. From these products, liquid water path and ice water path are calculated. TCWret also uses profiles of air pressure, humidity and temperature from measurements with Vaisala RS92 radiosondes and information about cloud height from measurements of the ceilometer CL51, which is on board the RV *Polarstern*.

During the measurement campaign, a data set with 5564 retrievals was created. A comparison to the simultaneously performed retrievals of the Cloudnet network on the *Polarstern* shows that:

- The LWP of both data sets are correlated. From this is concluded, that the retrieved LWP from TCWret is reliable. In addition, it could be shown using the TCWret dataset that during this measurement campaign also the measurement data of thin clouds (LWP $< 20\,\mathrm{g}\cdot\mathrm{m}^{-2}$) of the Cloudnet retrieval are reliable despite the given error of $20\,\mathrm{g}\cdot\mathrm{m}^{-2}$.

- As well as for the LWP, a correlation for $r_{liq}$ is observed. However, there is a increasing bias with increasing $r_{liq}$. This can be corrected using the results from Cloudnet.

- Only a low correlation can be found for the IWP, while $r_{ice}$ does not correlate. Therefore the IWP is considered to be less reliable than the liquid water-products.

Despite the difficulty in determining IWP and $r_{ice}$, this presented data set is useful for downward cloud radiative flux calculations. Since TCWret determines the cloud parameters from the spectral radiance, the calculated cloud parameters are those that match the observed radiance. This is also true if IWP and $r_{ice}$ are affected by errors.

In summary, the dataset of cloud parameters and water paths from TCWret provides a helpful complement to the results of the LWP from Cloudnet, but at the same time benefits from its $r_{liq}$. Due to the consistent calculation of cloud parameters over the entire cruise, the results from TCWret additionally provide information about clouds during PS107, where only EM-FTIR measurements are available.

## Appendix A:  Brief description of the Cloudnet synergistic retrieval

The LWP is determined using the HATPRO MWR, which uses two frequency bands between 22.24 GHz and 31.4 GHz and between 51.0 GHz and 58.0 GHz. A statistical retrieval has been set up using radiosonde data from Ny-Alesund, consistent with the procedure described in Löhnert and Crewell (2003) and leading a to a RMSE of $22.4\,\mathrm{g}\cdot\mathrm{m}^{-2}$. If a data point was classified as pure liquid, the effective radius of the cloud droplets is determined from the radar reflectivity and the LWP according to the retrieval of Frisch et al. (2002). The IWC was determined according to Hogan et al. (2006) via an empirical formula from

temperature and radar reflectivity. The IWP was determined by vertical integration of the IWC. The calculation of the IWP was carried out specifically for this study. The determination of $r_{ice}$ is done analogously to the IWC from the radar reflectivity and the temperature by an empirical formula (Griesche et al., 2020f).

## Appendix B:  Description of TCWret

### B1    Working principle of TCWret

TCWret retrieves optical depths of liquid water and ice water and the effective radii of liquid water droplets and ice crystals from infrared spectral radiances. The retrieval of microphysical cloud parameters is a nonlinear problem, so an iterative algorithm is needed:

$$\boldsymbol{x}_{n+1} = \boldsymbol{x}_n + \boldsymbol{s}_n \tag{B1}$$

Here $x_n$ and $x_{n+1}$ are the state vectors containing cloud parameters of the $n$-th and $(n+1)$-th steps and $s_n$ is the modification of the cloud parameters during the $n$-th iteration. The state vector contains the optical depths and effective radii

$$x_n = \begin{pmatrix} \tau_{liq,n} \\ \tau_{ice,n} \\ r_{liq,n} \\ r_{ice,n} \end{pmatrix} \tag{B2}$$

The governing equation to determine $s_n$ is

$$\left(K_n^T S_y^{-1} K_n + S_a^{-1} + \mu^2 S_a^{-1}\right) s_n = K_n^T S_y^{-1} \left[y - F(x_n)\right] + S_a^{-1} \cdot (x_a - x_n) \tag{B3}$$

The quantities in the equation are the jacobian matrix $K = \left(\frac{\partial F(x_i)_j}{\partial x_i}\right)$, the inverse of the variance-covariance matrix $S_y^{-1}$, the a priori $x_a$ of the cloud parameters and the inverse covariance matrix of the a priori $S_a^{-1}$, the measured spectral radiances $y$, the calculated spectral radiances $F(x_n)$ and the Levenberg-Marquardt term $\mu^2 \cdot S_a^{-1}$.

The aim of the iterations is to minimize the cost function $\xi^2(x)$.

$$\xi^2(x_n) = \left[y - F(x_n)\right]^T S_y^{-1} \left[y - F(x_n)\right] + \left[x_a - x_n\right]^T S_a^{-1} \left[x_a - x_n\right] \tag{B4}$$

Convergence is reached, if the change of the cost function is below a given threshold, here set to $0.1\%$:

$$\frac{\xi^2(x_{n+1}) - \xi^2(x_n)}{\xi^2(x_{n+1})} < 0.001 \tag{B5}$$

However, convergence in the sense of the cost function does not necessarily mean that the fitted and measured spectrum match. For example, the step size parameter of the Levenberg-Marquardt method could be so large that the cost function changes little. Then the convergence criterion is fulfilled, but the fit does not agree with the measurement. To identify these cases, a reduced-$\chi^2$-test is performed. This test is used to calculate the distance between calculated and measured radiance, taking into account the variance of the spectrum $\sigma^2$. It is defined as

$$\chi^2_{reduced} = \frac{1}{DOF} \cdot \sum_{m=1}^{N} \frac{y(\bar{\nu}_m) - F(x, \bar{\nu}_m)}{\sigma^2} \tag{B6}$$

with $DOF$ = number of datapoints - number of parameters. The microwindow is denoted as $\bar{\nu}_m$. As empirical values, we assume that all retrievals with $\xi^2_{reduced} < 1.0$ converged correctly. Results with $\tau_{liq} + \tau_{ice} > 6$ are excluded.

As we do not necessarily have prior information about the optical depths and effective radii, we decided to set the covariance of the a priori to large values. This shall ensure that the chosen a priori does not constrain the retrieval too strong. Initial values and a priori are set to equal values: $x_a = (0.25, 0.25, \log(5.0), \log(20.0))$. The logarithm was chosen so that all entries of $x_a$ have similar size. The variance-covariance matrix of the a priori is set to

$$S_a^{-1} = \begin{pmatrix} 0.04 & 0 & 0 & 0 \\ 0 & 0.04 & 0 & 0 \\ 0 & 0 & 0.047 & 0 \\ 0 & 0 & 0 & 0.047 \end{pmatrix} \tag{B7}$$

The values in $\boldsymbol{x_a}$ and $\mathbf{S_a}^{-1}$ are chosen empirically. Since initially no information about the cloud parameters is available, $\boldsymbol{x_a}$

and $\mathbf{S_a}^{-1}$ should not restrict the retrieval too much. Therefore, the variances in $\mathbf{S_a}^{-1}$ are set to large values.

Variances in $\mathbf{S_y}^{-1}$ are calculated from the spectral region between $1925\,\mathrm{cm}^{-1}$ and $2000\,\mathrm{cm}^{-1}$, where no signal from the atmosphere is expected. The variance-covariance matrix is assumed to be diagonal: $\mathbf{S_y} = \sigma^2 \mathbf{I}$. It is assumed to be the variance of the scene. To retrieve cloud parameters, only radiance from spectral intervals given in table (B1) is used. The variances of $\mathbf{S_y}$ propagated into the covariance matrix $\mathbf{S_r}$ of the result by applying a *transfer matrix* $\mathbf{T}$. In each step $\mathbf{T}$ is calculated taking

into account the current step size parameter $\mu$ by

$$
\begin{cases}
\mathbf{T_0} = \mathbf{0} \\
\mathbf{T}_{i+1} = \mathbf{G}_i + \left( \mathbf{I} - \mathbf{G}_i \mathbf{K}_i - \mathbf{M}_i \mathbf{S}_a^{-1} \right) \mathbf{T}_i
\end{cases}
\tag{B8}
$$

with $\mathbf{0}$ as zero matrix and $\mathbf{I}$ as identity matrix. $\mathbf{M}_i$ is the inverse of the term in the brackets on the left side of (B3) and $\mathbf{G}_i = \mathbf{M}_i \mathbf{K}_i^T \mathbf{S}_y^{-1}$. Diagonal elements of $\mathbf{S}_r$ are the variances of the final cloud parameters.

## Appendix C: Retrieval performance on simulated spectra

A set of simulated testcases containing spectral radiances of artificial clouds with known cloud parameters, created by Cox et al. (2016), will be used to test the ability of TCWret to retrieve $\tau_{liq}$, $\tau_{ice}$, $r_{liq}$ and $r_{ice}$. Additionally, the derived quantities LWP and IWP are discussed. This dataset contains several representative cases of Arctic clouds. Clouds are set to be either vertically homogeneous, topped by a layer of liquid water or with thin boundaries. Ice crystal shapes are mostly set

to be spheres, but some cases where calculated with hollow columns, solid columns, bullet rosettes or plates. All spectra are convoluted with a sinc-function to the resolution of the IFS 55 Equinox $(0.3\,\mathrm{cm}^{-1})$ and perturbed by a Gaussian distributed noise of $1\,\mathrm{mW} \cdot (\mathrm{sr} \cdot \mathrm{cm}^{-1} \cdot \mathrm{m}^{-2})^{-1}$: We modified the spectral radiance at each wavenumber by drawing a random number from a normal distribution with the true spectral radiance as mean of the distribution and $1\,\mathrm{mW} \cdot (\mathrm{sr} \cdot \mathrm{cm}^{-1} \cdot \mathrm{m}^{-2})^{-1}$ as its standard deviation. This value has been chosen, because it is near the observed standard deviation of the real spectra from the

measurement campaign of $0.82\,\mathrm{mW} \cdot (\mathrm{sr} \cdot \mathrm{cm}^{-1} \cdot \mathrm{m}^{-2})^{-1}$. Ice crystals are chosen to be spheres, thus only the testcases which are calculated with spherical ice crystals are used here. The influence of the chosen ice particle form will be adressed later.

Table (3) gives the correlation coefficients, mean biases and standard deviations between the retrieved cloud parameters of the testcases and the true cloud parameters. Additionally, the standard deviations calculated via the variance-covariance matrix is given. TCWret is able to determine optical depths and effective radii of the simulated spectra.

Of all direct retrieval products, the optical depths $\tau_{liq}$ and $\tau_{ice}$ have the highest agreement to the true cloud parameters. For the liquid phase, the difference to the true optical depths is $(-0.1 \pm 0.5)$. For the optical depth of the ice phase, the difference is larger with $(0.2 \pm 0.6)$. Since $\tau_{liq}$ and $\tau_{ice}$ include both optical depths and phase, the optical depth of the condensed water $\tau_{cw} = \tau_{liq} + \tau_{ice}$ as well as the fraction of ice in the optical depth $f_{ice} = \tau_{ice} \cdot \tau_{cw}^{(-1)}$ are calculated. Here it becomes clear that the optical depth can be determined accurately $(|r| = 0.99$, mean bias and RMSE $(0.1 \pm 0.2))$. It then also follows that the

deviations of $\tau_{liq}$ and $\tau_{ice}$ come from the phase determination. The deviation for the phase is $(0.1 \pm 0.3)$ with a correlation coefficient of $|r| = 0.70$.

When considering the effective radii, only results of $r_{liq}$ were used in where $f_{ice}$ is less than 0.9. For $r_{ice}$ only results with $f_{ice} > 0.1$ are considered. The mean difference of the retrieval from the true parameters and the root-mean-square error are $(-2.4 \pm 4.1)$ for $r_{liq}$ and $(3.0 \pm 10.0)$ for $r_{ice}$.

To estimate the influence of the A priori on the calculated result, the Averaging Kernel Matrix is used. The mean averaging kernel matrix over all retrievals is

$$\mathbf{A} = \begin{pmatrix} 0.87 & 0.09 & -0.15 & -0.09 \\ 0.11 & 0.90 & 0.19 & 0.03 \\ -0.04 & 0.07 & 0.50 & 0.05 \\ -0.16 & 0.05 & 0.03 & 0.42 \end{pmatrix} \tag{C1}$$

From equation (8) can be seen that the diagonal elements show for each parameter how strong the retrieved parameter is influenced by the a priori. Whereas the diagonal elements of the optical depths are near 1, indicating independence from the a priori, results for $r_{liq}$ and $r_{ice}$ show a larger influence from the a priori. From the trace of the averaging kernels follow 2.69 degrees of freedom of the signal.

The water paths are calculated from the optical depths and effective radii, therefore both quantities are influenced by the phase determination, as seen before in $\tau_{liq,ice}$ and $r_{liq,ice}$. The difference from the testcases is $(-1.6 \pm 6.3)$ for the LWP and $(1.9 \pm 10.0)$ for the IWP. However, the RMSE for the LWP is less than the minimum RMSE observed for LWP from microwave radiometer of at least $15\,\mathrm{g} \cdot \mathrm{m}^{-2}$ (Löhnert and Crewell, 2003).

Standard devations given by the variance-covariance matrix of the retrieval are shown in table (3) and named as ERR(OE). ERR(OE) is below RMSE for $\tau_{liq,ice}$, $r_{liq,ice}$, LWP and IWP. This might be due to uncertainties from the forward model - which are neglected here - propagated into the retrievals or due to the assumption of a diagonal variance matrix $\mathbf{S_y}$. To compensate these effects, the uncertainties from the posterior covariance matrix are scaled by RMSE/ERR(OE) with the RMSE from table (3) for the discussion in section 6.

## C1 Mean Bias and RMSE of effective radii

In the previous section, the results for $r_{liq}$ and $r_{ice}$ were only considered for a certain range of $f_{ice}$. Thus, liquid drops were only included in the consideration if the ice content was not higher than $90\%$. For ice crystals, the limit was at least $10\%$ ice content. In the following, these limits are shifted so that the results go in the direction of a single-phase retrieval for liquid water and ice.

Table (C1) shows the results for liquid water. The entries at the top describe cases with a higher proportion of liquid water than the cases at the bottom, which allow a higher proportion of ice. They are cumulative, which means that each record also contains the data of the record above it. From the testcases it follows that the RMSE becomes lower the fewer ice crystals are present. Also, the absolute mean bias decreases with lower ice content up to an ice content between $10\%$ and $30\%$. These results indicate that the presence of ice crystals lead to an underestimation of $r_{liq}$ by TCWret.

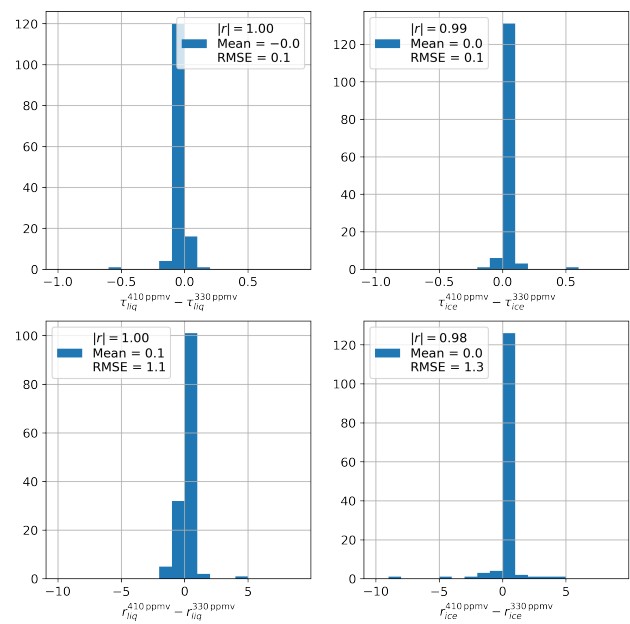

**Figure D1.** Histograms of differences for $CO_2$ concentrations of $410\,ppm$ and $330\,ppm$ for $\tau_{liq}$, $\tau_{ice}$, $r_{liq}$ and $r_{ice}$.

Table (C2) show the results for ice crystals. Here we introduced $f_{liq}$, which is defined as $f_{liq} = 1 - f_{ice}$ to create a table consistent with table (C1). Here one can see that the RMSE of $r_{ice}$ is almost independent of the water content. However, there is a dependence of the mean bias on water content. While removing clouds with very high water content leads to a decrease in absolute mean bias, the absolute value of mean bias increases for clouds with high ice content, so that TCWret underestimates
$r_{ice}$ of the simulated spectra.

## Appendix D: Influence of trace gas concentrations on the retrieval

In LBLRTM, a standard atmosphere was used for gases except water vapour. Therefore, the concentration of $CO_2$ is set to $330\,ppm$, although the real concentration in summer 2017 is about $410\,ppm$. To investigate the influence of an incorrect trace gas concentration, retrievals from the 11th June 2016 have been performed with both atmospheric concentrations of $CO_2$.
Differences are calculated for the cloud parameters $\tau_{liq}$, $\tau_{ice}$, $r_{liq}$ and $r_{ice}$ and shown in figure (D1). For all parameters, correlation coefficients between $|r| = 0.98$ and $|r| = 1.00$ can be observed. The maximum mean bias is observed for $r_{liq}$ ($0.1\,\mu m$) and the maximum RMSE is observed for $r_{ice}$ ($1.3\,\mu m$). From this it can be concluded that the influence of the trace gas concentration is negligible compared to the other uncertainties.

## Appendix E: Ice crystal shapes in the netCDF-file

Table (E1) refers to each key in the field *ice_shape* in the netCDF-file the corresponding ice crystal shape.

*Author contributions.*  PR performed measurements during PS106 and PS107, implemented TCWret and retrieved from infrared spectra. MP designed and built the measurement setup, performed measurements during the PS106.1, measured the emissivity of the blackbody radiator and gave advice in the development of TCWret. CW performed measurements during the PS106.2 and built the measurement setup. HG performed Cloudnet retrievals and gave advice in using the Cloudnet data. PMR gave advice in the application of the testcases. JN gave
advice in the setup of the measurement and the development of TCWret. All authors contributed to manuscript revisions.

*Competing interests.*  The authors declare no competing interests.

*Acknowledgements.*  We gratefully acknowledge funding from the Deutsche Forschungsgemeinschaft (DFG, German Research Foundation, TRR 172) - Projektnummer 268020496 - within the Transregional Collaborative Research Center - ArctiC Amplification: Climate Relevant Atmospheric and SurfaCe Processes, and Feedback Mechanisms (AC)3 - in subproject B06 and E02. We thank the Alfred-Wegener-Institute
and RV *Polarstern* crew and captain for their support (AWI_PS106_00 and AWI_PS107_00). The computations were performed on the HPC cluster Aether at the University of Bremen, financed by DFG within the scope of the Excellence Initiative.

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

**Table 7.** Datasets used in this publication.

| Description | DOI | Citation |
|---|---|---|
| Microphysical Cloud Parameters from TCWret | https://doi.pangaea.de/10.1594/PANGAEA.933829 | Richter et al. (2021) |
| OCEANET-ATMOSPHERE Microwave Radiometer HATPRO during PS106 | https://doi.org/10.1594/PANGAEA.919359 | Griesche et al. (2020a) |
| Cloudnet IWC during PS106 | https://doi.org/10.1594/PANGAEA.919452 | Griesche et al. (2020b) |
| Cloudnet $r_{ice}$ during PS106 | https://doi.org/10.1594/PANGAEA.919386 | Griesche et al. (2020c) |
| Cloudnet LWC during PS106 | https://doi.org/10.1594/PANGAEA.919383 | Griesche et al. (2020d) |
| Cloudnet $r_{liq}$ during PS106 | https://doi.org/10.1594/PANGAEA.919399 | Griesche et al. (2020e) |
| Ceilometer CL51 raw data measured during POLARSTERN cruise PS106.1 | https://doi.org/10.1594/PANGAEA.883320 | Schmithüsen (2017a) |
| Ceilometer CL51 raw data measured during POLARSTERN cruise PS106.2 | https://doi.org/10.1594/PANGAEA.883322 | Schmithüsen (2017b) |
| Ceilometer CL51 raw data measured during POLARSTERN cruise PS107 | https://doi.org/10.1594/PANGAEA.883323 | Schmithüsen (2017c) |
| Upper air soundings during POLARSTERN cruise PS106.1 (ARK-XXXI/1.1) | https://doi.org/10.1594/PANGAEA.882736 | Schmithüsen (2017d) |
| Upper air soundings during POLARSTERN cruise PS106.2 (ARK-XXXI/1.2) | https://doi.org/10.1594/PANGAEA.882743 | Schmithüsen (2017e) |
| Upper air soundings during POLARSTERN cruise PS107 (ARK-XXXI/2) | https://doi.org/10.1594/PANGAEA.882789 | Schmithüsen (2017f) |

**Table B1.** Microwindows used in TCWret to retrieve the microphysical cloud parameters of this dataset.

| Interval (cm$^{-1}$) |
| --- |
| $558.5 - 562.0$ |
| $571.0 - 574.0$ |
| $785.9 - 790.7$ |
| $809.5 - 813.5$ |
| $815.3 - 824.4$ |
| $828.3 - 834.6$ |
| $842.8 - 848.1$ |
| $860.1 - 864.0$ |
| $872.2 - 877.5$ |
| $891.9 - 895.8$ |
| $898.2 - 905.4$ |
| $929.6 - 939.7$ |
| $959.9 - 964.3$ |
| $985.0 - 991.5$ |
| $1092.2 - 1098.1$ |
| $1113.3 - 1116.6$ |
| $1124.4 - 1132.6$ |
| $1142.2 - 1148.0$ |
| $1155.2 - 1163.4$ |

**Table C1.** Determination of $r_{liq}$ depending on the cloud phase.

| Maximum $f_{ice}$ | $|r|$ | Mean Bias | RMSE | Datapoints |
|---|---|---|---|---|
| 0.1 | 0.97 | 0.6 μm | 2.1 μm | 19 |
| 0.3 | 0.90 | −0.2 μm | 2.3 μm | 38 |
| 0.5 | 0.79 | −1.0 μm | 2.7 μm | 87 |
| 0.7 | 0.70 | −1.6 μm | 3.2 μm | 151 |
| 0.9 | 0.59 | −2.4 μm | 4.1 μm | 192 |

**Table C2.** Determination of $r_{ice}$ depending on the cloud phase.

| Maximum $f_{liq}$ | $|r|$ | Mean Bias | RMSE | Datapoints |
|---|---|---|---|---|
| 0.1 | 0.61 | −5.0 μm | 12.0 μm | 31 |
| 0.3 | 0.59 | −0.3 μm | 12.0 μm | 80 |
| 0.5 | 0.67 | 1.3 μm | 10.2 μm | 141 |
| 0.7 | 0.65 | 2.6 μm | 10.2 μm | 180 |
| 0.9 | 0.65 | 2.9 μm | 10.1 μm | 193 |

**Table E1.** Ice crystal shapes in the netCDF-file and the corresponding number

| Key | Shape |
|---|---|
| 0 | Aggregates |
| 1 | Droxtals |
| 2 | Solid Columns |
| 3 | Hollow Columns |
| 4 | Spheroids |
| 5 | Plates |
| 6 | Bullet Rosettes |
| 7 | Spheres |