# Peer review of "A dataset of microphysical cloud parameters, retrieved from Emission-FTIR spectra measured in Arctic summer 2017"

_Earth System Science Data, 2021_

## Author Comment (AC1)

**1/ The word « significant » is used a lot of times along the paper to say « large », forgetting the quantitative, scientific meaning of that word in a statistical sense. Which hypothesis is tested to confirm that this is really significant ? To which null hypothesis does the p-value refer to ?**

The relates "significant" to the Pearson correlation $|r|$ coefficient. Whenever the p-Value is below 0.05, we assumed a significant correlation The p-value is interpreted here as a measure that two uncorrelated variables have the same correlation coefficient as the data sets for which $|r|$ is calculated.
To make it clear that we only calculated $|r|$ and did not perform a different test, we will remove the term "significant" from the manuscript. We will also remove the p-value from the manuscript and focus on the mean bias and the root-mean-square error in the analysis.

**2/ The authors never explain which variable has been calculated when they mention « significant correlations ». Does it refer to the Pearson correlation coefficient ? The coefficient of determination $R^2$ ? The Spearman's rank correlation coefficient ? In addition, providing « correlations », even though they are large, does not say anything about the discrepancies, but just mean than the parameters vary together. What are the biases and the root-mean-square errors ?**

The term "significant correlation" refers to the Pearson correlation coefficient.
We have used the correlation coefficient to test whether TCWret is able to determine optical depths, effective radii and water paths. Cloudnet serves as a reference here. Therefore, we mention it for in the tables. We still consider this necessary, as can be seen in the example of $r\_ice$. In addition, we have calculated the mean bias and the standard deviation (we will discuss the root-mean-square error in the next comment). In our opinion, however, these quantities are only of interest for correlated data.

Mean bias is indicated in the manuscript by the term "Mean". We will change the wording in the revised manuscript
.
**3/ There is a confusion about the term « standard deviation » that is used along the text (especially in Sect. 5.5) to express the RMSE.**

**The authors do not use standard quantitative scores widely used by the scientific community to evaluate the performance of an algorithm. What is called 'Mean' seems to be the 'Mean Bias'. This mean bias can be close to 0 due to compensation errors. The RMSE (root mean square error) usually gives a complementary information about the evaluation of performance. But what the authors use here, and that is called « STD (TC) », does not actually represent the full discrepancy between the retrieval and the true parameter as the RMSE would do. What has been calculated in the paper is the STD of the differences between the retrieval (r_i) and true parameter (t_i), which is :**

**STD(TC) = \sqrt{\frac{1}{n} \sum ((x_i - \bar{x})²)}**

**where x_i = r_i – t_i, and \bar{x} the average value of the x_i.**

**What should have been calculated is rather :**

**RMSE = \sqrt{\frac{1}{n} \sum ((r_i – t_i)^2)}**

**which would automatically provide larger values than the « STD(TC) » used here.**

**How much is the RMSE for each retrieved parameter ?**

We have given the standard deviation instead of the RMSE in the manuscript. In this way we wanted to give a comparability to the retrieval error, which in TCWret comes from the covariance matrix and is given here as standard deviation.

However, the magnitudes of RMSE and STD(TC) are comparable. For the LWP of TCWret and Cloudnet, STD(TC) = 10.13 gm-2 and RMSE = 10.43 gm-2. A larger change is only to be considered for the effective radii. Thus, for r_liq STD(TC) = 2.95 um and RMSE = 5.26 um.

In the revised manuscript, we will use RMSE instead of STD(TC).

**4/ Standard deviations are given with an accuracy of 2 significant digits after comma, for instance in the abstract. Is it really realistic ?**

**If I understand what has been calculated, the standard deviations are only dispersions. Did the authors also calculate the uncertainties on the retrieved parameters ? This is a crucial information for the reader interested in using this dataset.**

The size of the error bars in Figures 10 to 12 is based on the retrieval errors. Here the retrieval error has been scaled with the size STD(TC) to account for the larger inaccuracies found in the test cases. The retrieval errors are also given in the published data set.

As a basis for the chosen two significant digits we used the step size of the retrieval (1e-3) and the interval in which the r_eff are entered in the single-scattering databases (5e-1 to 1). Since we are discussing mean values, we allow one more significant digit, so that we have at least one and at most two significant digits.

However, due to the interval of r_eff, it is reasonable to give only 1 significant digit.

**5/ The methodology is justified in a weird way (e.g. L 41) : there are plenty of algorithms based on a similar approach that are freely available. Some of them are actually mentioned later in the paper (MIXCRA, CLARRA, XTRA). Can the authors explain exactly what is new in comparison to other published algorithms ?**

The algorithm itself is not new. However, it was not possible to use CLARRA and XTRA because CLARRA had not yet been published at the time of the measurements and the intended evaluation of the measurements.
XTRA was no longer available and it used a self-developed radiative transfer model instead of DISORT. MIXCRA was not freely available. Only the LBLDIS model was freely available. Because of this, we decided to develop our own retrieval, but based on LBLDIS.

The focus of this publication is on the data, rather than on the retrieval algorithm. To make this clear, we have shortened the section in which we describe and test the retrieval and moved the large part to the appendix.

Specific comments :

**L 10-12 : it is not clear in the abstract what is the reference dataset and which one is evaluated in the paper. This sentence gives the impression that the authors aim to evaluate the data on opticall thin clouds measured bu microwave radiometers withing the Cloudnet framework (not from FTIR spectrometer).**

We evaluate the TCWret dataset and use the Cloudnet dataset as a reference. We will formulate this more clearly in the revised manuscript.

Since in the case of thin clouds (LWP < 20 gm-2) a lower RMSE than expected is observed, and we also evaluate TCWret independently of Cloudnet using simulated spectra, we conclude that TCWret can be used in this case to make statements about the Cloudnet dataset.

**L 13 : The syntax used here (« allows to perform[…], which was the case[...] ») is misleading. The calculations of the cloud radiative effects are not performed in this study.**

With this statement we want to show a possible use of this data set and thus illustrate the relevance of the data presented. We will reformulate this sentence in the revised manuscript

**L 37 : « smaller uncertainty » : Based on the scientific litterature, how much is it ?**

The statement is mainly based on the comparison with the simulated spectra, according to which we have an RMSE at the LWP of 6.3gm-2. Also Turner (2005) found a RMSE of 5gm-2 using MIXCRA in comparison to other retrieval techniques like radar, statistical and physical retrievals.

However, we decide to reformulate this and cite Turner et al. (2007), who gives an upper threshold for the LWP, which can be retrieved using a FTIR spectrometer.

**Fig. 1 : The ship track is not mentioned in the figure caption.**

We added the ships track to the caption.

**L 102 : « accuracy of ≥ ± 5 m ». This is confusing. Does it mean that the absolute error is larger than 5 m ?**

The error of the ceilometer is +- 1% of the determined ceilings, but at least +-5m. We have changed the description accordingly.

**L 105 : Do the data from the Vaisala ceilometer and the Cloudnet profiles at least agree for the P106 period ? It is important to give the bias here as the ceilometer data are used during the entire cruise.**

There is a mean bias between the cloud base height stated by Cloudnet and the ceilometer of -639m (median bias of -47m), which means on average the Cloudnet cloud base height is larger than the ceiling given by the ceilometer, and a root-mean-square error of 1870m. Since Ceilometer and Cloudnet have different measurement intervals, we have averaged to 5 minutes.

Nevertheless, we decided to use the Ceilometer data because in any case we have to use it for the PS107 and thus get a consistent dataset.

**Sect. 5 is very long. It gives the impression that the paper focuses on the presentation of an algorithm rather than on the description and evaluation of the EM-FTIR measurements. Can the authors comment on the main objective of this paper ?**

The focus is on the data. However, since Cloudnet using the HATPRO for LWP has a lower sensitivity to thin clouds, while TCWret can only be applied to thin clouds, we decided to include the section on retrieval performance. To make it clear that the paper is about the data, we have moved the characterisation section to the appendix and limit the main text to the products of TCWret and an error estimation
.

**L 116-118 : What are the main differences between the different algorithms ?**

All retrievals are physical retrievals. While MIXCRA uses LBLDIS to couple the models, CLARRA uses its own coupling algorithm. XTRA uses its own radiative transfer model and performs the retrieval on spectral radiances. MIXCRA performs the retrieval on the cloud emissivity.

However, since we have no information about the cloud profile during PS107, we apply retrieval to the radiance, as is done in XTRA.

**L 123 : Are aerosol optical properties included in the calculations, especially for dust particles in the infrared spectrum ?**

No, aerosol optical properties are not in included in the retrieval

**L 138 : What about the size distribution of ice crystals ? Is it also prescribed ?**

Yes, the size distribution of ice crystals is a gamma function, too

**Table 2 : Is this table really necessary ? The extreme values of the spectrum and the number of spatral bins may be enough here.**

We consider the table useful. With the exact information about the windows it can be checked that the retrieval was only applied where there is little influence of gases. We have used other publications as a guide, which also give the spectral position of the microwindows (Turner 2005, Rowe et al 2019).

**Eq. 7 : What does $\nu_n$ mean ? I had understood that $\nu$ was the mean wave number in each intervall. Why should it be a function of n, defined as a iteration step in Eq. 3 ?**

The n denotes the number of the microwindow and is a different index as in equation 3. We rename it to m in equation 7 in the revised manuscript.

**Eq. 8 : Where do these values come from ? Have the authors perform a sensitivity study to evaluate the influence of $S_a^{-1}$ on the final retrieved parameters ?**

Since we wanted to keep the retrieval independent of Cloudnet and have no other information about the cloud parameters that can be used in $x_a$, we decided to insert high values for the variances in $S_a$ (sigma_tau = 5 and sigma_r = 100 um). This is to ensure that the calculated result is not influenced by the chosen A priori and is in line with Turner's (2005) approach.

We did not conduct a sensitivity study for different variances in $S_a$.

**L177-178 : It would better to use \sigma_{ice} everywhere, rather than ext(rice). The extinction coefficient of ice crystals should also depend on the temperature as the refractive indices do.**

ext(ice) is used in the SSP databases, therefore we have mentioned it here. We used the SSP databases that come with LBLDIS. Since there are no temperature-dependent refractive indices for ice crystals in the databases, we have not entered the temperature dependence in the equations.

**L 205 : As a consequence, the variance of rice is written by this convention \sigma_{rice}. To avoid confusion with the extinction coefficient of ice crystals, the authors may want to note this latter differently, for example \alpha_{ice}(r_{ice}).**

We changes the extinction coefficient to alpha and the extinction cross section to beta.

**Table 3 : What does the « maximum m testcases » mean ? It has not been defined here.**

This is the maximum value for the respective quantity found in the test cases. Only the successful retrievals are taken into account

For example, the largest r_liq in the successfully retrieved test cases is 22um, the largest LWP is 46.90gm-2.

**L 220 : « Significant » does not mean « large », but has a precise statistical meaning. To confirm that a correlation is significant, the authors must perform a statistical test and provide the values of the result of this test.**

Please see comments above. We use the term to refer to the p-value of the Pearson correlation coefficient, which is below 0.05.

**L 220-221 : I am not sure if I correctly understand this sentence. What are the given uncertainties ?**

By this we meant that the variables can be determined taking into account the standard deviations STD(TC). Weh ave reformulated this sentence in the revised manuscript.

**L 229-230 : How much are the results sensitive to the choice of the threshold of f_{ice} ? If we choose thresholds at 0.8 / 0.2, are the results significantly different ?**

Changing the thresholds from 0.9 to 0.8 for liquid droplets removes those clouds from the evaluation, which contain the least fration of liquid water. We found that this decreases the mean bias and RMSE for r_liq. The RMSE of r_ice is independent of the choice of ice fraction. However, the absolute mean bias decreases when the maximum water content of the clouds is reduced. This is true up to a maximum water content of about 0.3, after which the absolute mean bias increases.

We added a section in the appendix to address this question.

**L 233 : I don't get this point. Here, \bar{r} has been calculated from the knowledge of r_{liq}, r_{ice} and f_{ice}. How can it be « estimated independently » ? Do the authors rather want to say that \bar{r} results from a compensation of errors in the cloud parameters used for its calculation ?**

That is true. The statement refers to the compensating errors of r_liq, r_ice and f_ice, which become clear by introducing bar{r}.

**L 237 : This should be said before when A is introduced for the first time.**

We have placed the explanation under the equation that describes the AVK matrix for the first time.

**L 244 : Are the authors comparing the same variables (« called standard deviations ») that what is used in the litterature (Löhnert and Crewell, 2013) ?**

Löhnert and Crewell refer to the RMSE, while we refer to STD(TC). We will change this. However, since STD(TC) and RMSE are similar for the liquid water pathway in TCWret, the associated statement does not change.

**Fig. 4 : This caption is not very explicit ? What is represented exactly ?**

Here we show the total error, which we represent as the sum of instrument error and "interpolation error".
The interpolation error follows from the comparison between the linear interpolation between two radiosonde measurements and the ERA5 atmosphere at the position of the measurements. We query the ERA5 atmosphere for each hour. Then we calculate the atmospheric profiles from the radiosondes once per hour by linear interpolation. We calculate the difference, average over one day and calculate the standard deviation.

**L 274 : The parameter « h » has not been defined. Are « h » and « \Delta \epsilon » equal ?**

h denotes the step size which we have used for the numerical calculation of the partial derivative. We added an explanation of h.

**L 282-283 : Please comment those values. They seem extremely large to me. Does it suggest that the effective radii and liquid/ice water contents cannot be estimated by this approach ?**

The values in Table 4, which represent the partial derivatives, are erroneously standard deviations instead of mean values. Using the standard deviation leads to wrong partial derivatives (deviation from the partial derivative instead of the partial derivative itself). Using the mean values, the errors decrease to tau_liq = 0.4, tau_ice = 0.3, r_liq = 3.3, r_ice = 13.1, LWP = 2.8 and IWP = 5.6

**L 286 : What do the authors mean by the « standard deviations of r_{ice} » ? Is it a std on the parameter « r_{ice} » or the std on a difference as it is the case along the paper ?**

The latter is true, it is the standard deviation on a difference.

**L 290-291 : This turns out to be only a partial conclusion. In the case of hollow columns for example, the retrieval is particularly bad in almost half of the cases, but it is not mentioned here.**

There is a formatting error in table 5 . The 10.72, which are at hollow columns vs. spheres, belong to plates vs. spheres. This means that hollow columns are not worse than the others, with the exception of bullet rosettes and plates.

**L 294 : What are « differentials of IWP » ? Are they simply differences ?**

Differences are meant by this expression.

**Tables 5, 6, 7 : « Difference of r_{ice}/IWP/ \tau_{ice} ». What are the reference parameters ?**

Reference parameters are the r_ice/IWP/tau_ice of a different shape. We For example, the 6.76 in table 5 (first row, second column) is the mean difference between spheres and aggregates.

However, following the revised evaluation of r_ice, we decided to remove the section about the ice crystal shape, as it does not add any further value to the analysis.

**Fig 5 : Are the histograms normalized by the total number of occurrences ? And also by the width of the bars/intervalls on \tau ?**

The histograms are normalised so that the integral of all bars equals 1. We will therefore replace them so that the representation is consistent with the rest of the histograms.

**Fig 5 : The authors said before that the algorithm was not used when the total optical depth of the cloud was lower than 6. Why are there values for \tau_{liq} > 6 ? It such values are removed from the analysis, how are the results modified ?**

This plot is all results, including those discarded because of the limit tau > 6. We will exchange the histogram. However, those values do not appear in the analysis

**Fig. 6 : It seems that in 2000 cases, there is no IWC. Does it mean that there are 1000 occurrences of pure liquid clouds ?**

That is correct.

**Fig 7 : How many cases correspond to the criteria set for the plot (optical depths > 0.1) ?**

That is an incorrect description. We are not dealing with optical depths above 0.1, but with an ice or water content above 10%. These are 4111 out of 4590 for r_liq and 2153 out of 4590 for r_ice. We have inserted these figures

**L 301 : This is not the place for this. It is said later in a specific section.**

We removed this sentence here.

**L 350 : Do the authors conclude that the geometry of ice crystals was incorrect ?**

Yes, we suspect that the ice crystal shape does not correspond to reality. This already follows from the fact that we can only calculate a single r_ice for the entire cloud and can only assume one ice crystal shape.

**L 354 and following : This is a very strange way to write differences between two datasets. In the litterature, when we write «  m ± s », it stands for a mean value m and a dispersion value, generally expressed by the standard deviation s. If we would rather to express a confidence interval around m, it is usually written m ± s/ \sqrt{n}, where n is the number of values in the dataset. When comparing two datasets, it is common to use the mean bias (MB) and the RMSE, but they are never written as MB ± RMSE has the second one does not stand for a dispersion around the first one. Both are statistical variables expressing the discrepancies between a model distribution and a reference or observed value. In this section and the next ones, the way the values are given is very confusing.**

We use Mean Bias +- Standard deviation in the manuscript. We will change this to Mean Bias +- RMSE in the revision.

**Fig. 10 : Values don't seem correlated and the r parameter is indeed very low. Are the data derived from TCWnet really reliable ?**

$r\_ice$ from TCWret does not correlate with the data from Cloudnet. Since IWP is calculated from $r\_ice$ and $tau\_ice$, IWP is also subject to corresponding uncertainties. It is conceivable that with a better estimation of $r\_ice$ (for example, by restricting the retrieval by an A priori), the IWP can also be determined more precisely. To answer the question conclusively, $tau\_ice$ would also have to be evaluated. However, this is not part of the Cloudnet product.

Furthermore, the observed IWP during the measurement campaign is at the lower limit of what TCWret can determine according to error estimation.

To investigate more precisely whether the IWP of TCWret is reliable, it would have to be applied to clouds with higher IWP than observed during the measurement campaign.

**L 358 : « means and standard deviations for LWP and r_{liq} are shown ». In Table 9 caption, the text seems to indicate that the given values are means and standard deviations of differences. Which one is correct ?**

These are differences. We have changed the wording in the caption.

**Sect. 6.1 : This small subsection is confusing and not very rigorous. Do the values given here significantly (e.g. in its statiscal sense, meaning using a statistical test) differ from the values obtained for the testcases ?**

In this section we want to present the data set and the magnitudes of the retrieved cloud parameters during the measurement campaign.

We did not perform a statistical test to check whether the measurements differ from the test cases. However, we believe that this is likely:
-    The test cases contain clouds that are typical for winter and summer. However, our measurements are only from summer
-    Some of the clouds in the test cases are much higher and much colder, while the temperatures of the measured clouds are in the 0°C range.

Alternative methods to the method used here with the testcases of Cox et al. are, in our opinion, the following:
-    Use only the part of the test cases that corresponds best to the measurements in terms of setup. The disadvantage is that this selection becomes small. However, out of a total of 252 successful retrievals, only 65 are from the period between May and mid-August.
-    Create your own tests and check the results. This would allow any number of data to be generated, but for the test of the retrieval, the retrieval itself would generate the spectra, which could leave systematic errors undetected.

After weighing up the pros and cons, we decided to use the testcases from Cox et al.

**L 318 : « there a less cases » : How many ? Which fraction does it represent ?**

We have 5564 retrievals in total. Of these, we classified 2158 as mixed-phase ($0.1 < f\_i < 0.9$), 2899 as single-phase liquid ($f\_i < 0.1$) and 507 as single-phase ice ($f\_i > 0.9$). We will add this data in the revised manuscript.

**Tables 8, 9 : Do «Mean» and « STD » stand for the mean and standard deviation of the parameter 'IWP', 'r_{ice}' or the standard deviation of the discrepancies between the variables retrieved from the TCWnet and Cloudnet ? In this latter case, it would be better to use the mean bias and the RMSE.**

Mean and STD stand for the discrepancies. Mean is already the mean bias and STD will be replaced.

**Tables 8, 9 : What has been tested exactly by the p-value (never mentioned in the text) ? To which null hypothesis does the statistical test correspond ? What do the authors conclude with such values ?**

See above. We use the Pearson correlation coefficient. The p-value is interpreted here as a measure that two uncorrelated variables have the same correlation coefficient as the data sets for which |r| is calculated. Our conclusion is that the retrievals from TCWret are reliable (p < 0.05) or not reliable (p < 0.05). Since the IWP is influenced by r_ice, its reliability is limited here.

This is particularly important for the PS107 data, as there is no Cloudnet data for comparison here.

**L 368 : « significant correlation » : the authors may rather want to say that the correlation coefficient is large enough. The statistical significance can then be discussed using the statistical test (and the associated p-value under a specified null hypothesis).**

We use the term "significant" here to refer to the p-value, which is smaller than 0.05. However, we have removed the term in the revised manuscript to make it clear that we are only talking about the Pearson correlation coefficient and not any other statistical test.

**L 405-406 : The error is as large as the threshold on LWP. Can we say something about the agreement of the two datasets in this case ?**

The idea behind this analysis is that the error in this case is less than 20gm-2. To confirm this, we want to investigate whether the data sets are correlated and how high the RMSE is. If one of the two retrievals cannot determine the LWP for clouds with LWP < 20gm-2, we expect no correlation.

**L 407-410 : No statistical test has been performed nor discussed. It is therefore impossible to say anything about the significance.**

We will reformulate this paragraph.

**L 409 : « too small », « overestimated » : this is very qualitative. By how much ? Are the differences larger than the uncertainties ?t**

The mean bias (-16.77) is greater than the standard deviation (12.83). That is what this statement referred to. However, this no longer applies to the RMSE (21.11). However, since the results do not correlate, we have reformulated the statement regarding r_ice in the revised manuscript.

**L 414-417 : The paper underlines that the results on r_{ice}, r_{liq} and IWP are different from those derived by Cloudnet. Is it worth publishing such results if the values significantly differ ? Which dataset is reliable ?**

Furthermore, a correlation of the TCWret and Cloudnet data sets can be observed. We have completed the interpretation and show that r_liq of TCWret does not correspond to the mean r_liq of Cloudnet. We conclude this because the mean bias and RMSE are lower compared to the maximum effective radius of a cloud in the Cloudnet data. The remaining quantities r_ice and IWP are part of the retrieval result and therefore we see it necessary to publish these results as well and put them in context with the Cloudnet data.

For the radiative flux calculations mentioned in the abstract, it is also not sufficient to publish only the LWP. Here, for the data from TCWret, clouds with the calculated parameters correspond to the observed radiative fluxes in the measurement range of the FTIR spectrometer. Thus, LWP, IWP, r_liq and r_ice should provide a reasonable estimate of the radiative flux in the long-wave spectral range, even if IWP and r_ice are subject to greater uncertainties.

**L 56-60 : Only 4 lines do not justify a whole section. Sections 2 and 3 should be combined.**

We combine these sections into one section in the revised manuscript.

**Technical comments :**

**The syntax is often incorrect and there are a lot a typos in the current version. The text needs to be checked very carefully, and ideally be corrected by a native speaker.**

**L 51 : A closing parenthesis is missing here.**
The parenthesis has been added

**L 55 : Replace « is provided » by « are provided ».**
Done

**Sect. 3 : The authors regularly switch from the present to the past tense and vice versa. Please keep only one.**
We rephrased it and keep the text in present

**L 64 : Replace « has » by « had »**
Done

**L 66 : Replace « has a movable mirror which gives » by « has a movable mirror giving ».**
Done

**Fig 3 : What does Emissivity (1) mean ? If « 1 » is only used to say that the emissivity is a dimensionless variable, it is better to remove it.**
Yes, the (1) was meant to say, that the emissivity is a dimensionless variable. We have removed it

**L 81 : Replace « of high temperature » by « at high temperature ».**
Done

**L 82 : interferograms**
Done

**L 83 : procedure**
Done

**Some acronyms are not defined in the text, e.g. OCEANET (L90-91), HATPRO (L. 94).**
We added the full name of HATPRO. OCEANET is the name and not an acronym

**L 101 : Replace « Informations … are » by « Information … is ».**
Done

**L 127 : Replace « An vertically inhomogenious » by « A vertically inhomogeneous ».**
Done

**L 131 : single-scattering albedo**
Done

**L 131 : different wavenumbers**
Done

**L 133 : Replace « temperature depended » by « temperature dependent ».**
Done

**L 138 : «were chosen in a way ».**

Done

**L 146 : « steps »**
Done

**L 149, 169, 184, 193, 322 : Please avoid starting a sentence by the final dot of the previous equation.**
Done

**L 150 : Replace « inverse covariances » by « inverse covariance matrix ».**
Done

**Eq. 6 : Remove the square on x_{n+1}**
Done

**Eq. 7 : x should be a vector, as defined by Eq. 3.**
Done

**L 162-163 : Correct as : « we assume that all retrievals […] correctly converged . »**
Done

**L 164 : « information ».**
Done

**L 166 : x should be replaced by x_a.**
Done

**L 177 : « extinction coefficient »**
Done

**L 210 : Replace « homogenous » by « homogeneous ».**
Done

**L 218 : « mean deviations » : do the authors use this term instead of the widely used « mean biases ».**
We changed the wording to Mean Bias

**L 219 : « true cloud parameters ».**
Done

**L 219 : « the standard deviations ».**
Done

**L 230 : there are two verbs in the sentence 'is' and 'are'. The sentence must be reformulated.**
Done

**L 238 : « retrieved »**
Done

**L 249 : Add a « that » : « so that it matches ».**
We rephased this sentence

**L 251 : « humidity ».**
Done

**L 274 and L 276 : Replace « differential quotient » by « partial derivative ».**
Done

**L 275 : Remove « as ».**
Done

**L 277-278 : Some parentheses are not at the right place or are missing in all expressions.**
We added the missing parentheses

**L 282 : Make two sentences here. « . This gives... ».**
The entire section has been modified

**Tables 5, 6, 7 : « bullet rosettes ».**
These tables have been removed in the revised manuscript

**Fig. 5, 6, 7 : Replace « plot » by « panel » in the figure captions.**
Done

**Fig 7 : Replace « distribuin » by « distribution ». Correct « the optical depths is » by « the optical depths are. »**
Done. Latter was meant to be the phase fraction. We corrected the sentence

**L 308 : Replace « is shown » by « are shown ».**
Done

**L 308-309 : « Similar for ... » : Please make a sentence.**
We have expanded the expression into a sentence

**Fig. 8 : Replace « Statistics » by « histogram ».**
Done

**Fig. 9 : Replace « divided by the chosen ice particle shape » by « for each ice particle shape ».**
Done

**L 327 : Replace « are the spectral windows » by « is the spectral window ».**
Done

**L 330 : « intransparent » : Do you want to say « opaque » ?**
We changed the wording to opaque

**L 334 : Replace « result » by « results ».**
Done
**L 334 : Replace « where » by « when ».**
The sentence has been rephrased.

**L 336 : bullet rosettes.**
Done

**L 336 : I see a small fraction of hollow columns, spheroïds and spheres. Have they been removed in this analysis ?**
We have modified the description of how we determined the ice crystal form

**L 338 : Add a « by » : « This is motivated by the following. ».**
Done

**L 338 : « The results of […] show that ».**
Done

**L 339 : « and \bar{r} can be seen that ».**
We removed bar{r} from the manuscript

**L 340 : « with a too small r_{ice} and a too large r_{liq}.**
Done

**L 361 : Replace « is » by « are ».**
Done

**L 363-364 : I can't understand this sentence. Please reformulate.**
Done

**L 380 : « r_{liq} thus improves » : this syntax is incorrect. The algorithm improves the retrieval of r_{liq}.**
That is correct. However, we removed this sentence from the manuscript

**L 382 : accessibility**
Done

**L 388 : Remove « in this publication ».**
Done

**L 403 : Add a « that » at the end of the sentence.**
Done

---

## Author Comment (AC2)

**My main concern with the manuscript is related to the comparisons to Cloudnet. First, the manuscript does not clearly describe the Cloudnet data. In the abstract, when referring to the Cloudnet data: "...liquid water path retrievals from microwave radiometer ..." (line 8) which suggests the LWP from Cloudnet is from solely the microwave data. On the other hand, the Cloudnet data is "combined cloud radar, lidar, and microwave radiometer" (line 5). Please clarify exactly how the Cloudnet works here - it is important to know how Cloudnet is retrieving the variables that are compared to the variables from the emission FTIR (IWP, LWP, particle size). Are the different variables simply retrieved from the individual remotely sensed measurements? How would that then work for particle size? Doesn't that require joint lidar/radar?**

The Cloudnet retrieval is described in Illingworth et al. (2007). A complete description of the cloudnet retrieval was not planned, so we have only provided a summary of the measurement equipment used.

The retrieval for this measurement campaign is described in Griesche et al. (2020) and we will give a brief description of the used quantities here and refer to this publication in the following.

The LWP is indeed determined from the measurements of the Humidity and Temperature Profiler( HATPRO). The frequency bands from 22.24 GHz to 31.4 GHz and from 51.0 GHz to 58.0 GHZ are used. A statistical retrieval is applied, which was set up using radiosonde data from Ny-Alesund. This retrieval corresponds to the retrieval described in Löhnert and Crewell (2003).

With the calculated LWP and radar reflectivity, $r\_liq$ is determined according to Frisch et al. (2002) for data points classified as liquid water.

We calculated the IWP ourselves via vertical integration of the IWC, which is determined by an empirical formula from temperature and radar reflectivity according to Hogan (2006).

$R\_ice$ is proportional to the ratio of IWC and the visible extinction coefficient and is thus also calculated using temperature and radar reflectivity.

IWC and $R\_ice$ is only calculated for data points classified as ice or supercooled water.

In principle, the LWC is also part of the cloudnet retrieval, which is determined on the basis of the LWP. However, since only the integrated quantities can be determined with the FTIR, the LWC is neglected here.

Before the individual retrievals, however, a classification of the data is performed (liquid, ice, aerosol, ...), which is made possible by the combined use of radar and lidar (Illingworth et al. 2007).

**A main conclusion of the manuscript is that the Cloudnet LWP measurements are more accurate than the 20 g/m2 uncertainty that is quoted on the product. The supporting evidence is that the LWP retrievals from both methods (cloudnet and TCWret) show high correlation even when LWP < 20 g/m2. I am not sure this follows - it depends on how the original 20 g/m2 uncertainty estimate was derived for the cloudnet data. If this was assessed by comparison to independent "truth" estimate, then if the true errors in both Cloudnet and TCWret are correlated, one would see this correlation even though the true error in both methods is still 20 g/m2. For example, the 'parameter' uncertainties discussed in section 5.6 could drive correlated errors in both retrievals.**

The data on which the retrievals are based - with the exception of the atmospheric profiles - are different. TCWret also does not use Cloudnet's retrieval results as a priori information, so an influence on the results is not to be expected here either. However, TCWret and Cloudnet use the same radiosonde measurements of temperature and humidity for their retrievals, so the possibility of correlated errors cannot be completely ruled out. However, since Cloudnet and TCWret use the same atmospheric profiles, inaccuracies in them should not matter, so we believe the parameter errors are also negligible. This leaves only the retrieval errors.

The 20 gm-2 is the RMSE of the statistical retrieval, while the uncertainty of TCWret is that from the covariance matrix of the optimal estimation.

Additionally, all readings are averaged over a 2 minute period for better comparability. Again, I see the possibility that this reduces Cloudnet's true uncertainty. However, similar results show when results are averaged over one minute.

**In section 5.5 (and 6.2), it would be much more informative to also show the posterior correlation matrix. An important point in the discussion section is the tradeoff between r_liquid and r_ice. It would be very useful to know if the output of the OE algorithm shows this correlation (e.g. the r_liquid - r_ice correlation term should be negative and have a large magnitude).**

For clarification: We understand the posterior correlation matrix to be the matrix that contains the mutual correlations, for example Corr(tau_liq, tau_ice).
We found that the Averaging Kernel Matrix (AVK) already contains all the necessary information about mutual dependencies and dependencies on the A prior. In addition, the AVK provides information about the degrees of freedom (tr(AVK) = degrees of freedom). Nevertheless, we will provide the Posterior Correlation Matrix in Section 6.2, which will be used in the interpretation of the results. In Section 5.5, which will be moved to the Appendix, we will continue to consider only the AVK.
The correlation coefficient Corr(r_liq, r_ice), however, is not negative and also has no larger magnitude than the other correlations. The posterior correlation matrix of the mixed-phase clouds of the measurement campaign is shown:

|       | t_liq     | t_ice     | r_liq     | r_ice     |
|-------|-----------|-----------|-----------|-----------|
| t_liq | 1.000000  | 0.503039  | -0.072331 | -0.409233 |
| t_ice | 0.503039  | 1.000000  | 0.016336  | -0.230436 |
| r_liq | -0.072331 | 0.016336  | 1.000000  | 0.130602  |
| r_ice | -0.409233 | -0.230436 | 0.130602  | 1.000000  |

Here, just as in the AVK from equation 22, one can see that there is a greater correlation between t_liq and t_ice, which is in agreement with the difficult phase determination. In addition, there is a high correlation between r_ice and tau_liq, which also suggests that these quantities cannot be determined independently or that in the clouds the optical thickness for liquid water is indeed correlated with the ice crystal radius

The correlation between r_liq and r_ice is low. We suspect that the expectation that the r_liquid - r_ice correlation term should be negative and have a large magnitude is based on the explanation in Section 6. However, since bar{r} also contains f_ice and thus also the optical depths, the correlation is not as direct as stated here. We will revise the explanation of the mean biases in this respect. We also have removed bar{r} from the manuscript as it does not add any further value to the interpretation.

**The parameter error discussion in section 5.6 has some unclear aspects. At line 257 "Each of these modifications is applied individually, creating three new datasets". If the modifications are made as described (e.g., add +1K to each cloud's temperature), it seems like this would only tend to create a mean bias in the retrieval, not increase the uncertainty. If it was done in this way, then these parameter errors would seem to be significantly underestimating the actual parameter error magnitude.**

The aim of this section was to carry out an error propagation analogous to equation 16, but this time taking into account variables that enter the retrieval as parameters. These are temperature of the cloud, humidity and the temperature of the blackbody, which influences the calibration and thus the size of the radiance itself. According to equation 16, to determine the uncertainty with respect to one of these quantities, I need the variance or standard deviation, as well as the partial derivative with respect to the quantity. The former are the specified instrument errors, while the latter is unknown,

since the retrieval by temperature. Humidity and radiance does not vary. Thus, the partial derivatives are determined separately in this case:

- We apply the described perturbation (e.g. add +1K to cloud temperature) to the test cases and calculate the cloud parameters under this perturbation.
- These results are compared with the test case retrievals without disturbance. I calculate the differences.
- The sigma (Table 4) is then the partial derivative by temperature, humidity and radiance normalised to unit size (1K, 1%rh and 1 RU). Since I want to calculate unperturbed partial derivatives, I can only apply one perturbation per data set. There is an error here because in the paper it is the standard deviation, although it must be the mean values.
- To calculate the total parameter error, Equation 18 then multiplies the unit errors by the partial derivatives.

Since we used these modifications to calculate the partial derivatives, which are weighted with the real instrument errors, this procedure should yield more than a mean bias.

**At line 280-283 at the end of the section, I believe the authors are attempting to combine the various error estimates into one final combined error, but I cannot follow the explanation. Where does Delta T = 2K, Delta q = 17.5% come from? How should the reader interpret these Deltas from the blackbody emissivity and temperature versus the radiance error att line 256? What are these final "deltas" supposed to represent? If these are supposed to be the combined parameter and calibration errors, these are much larger than the range of the OE errors as reported in Table 3.**

The Delta (Delta T, Delta q and Delta L) consist of

- The specified instrument errors of the radiosondes and the blackbody.
- What we have called here the interpolation error. This interpolation error takes into account the errors introduced into the temperature and humidity by linear interpolation between two radiosonde data setsThe interpolation error follows from the comparison between the linear interpolation between two radiosonde measurements and the ERA5 atmosphere at the position of the measurements. We query the ERA5 atmosphere for each hour. Then we calculate the atmospheric profiles from the radiosondes once per hour by linear interpolation. From this we calculate the difference, average over one day and calculate the standard deviation.

We added the two types of error and plotted them in Figure 4. Delta T = 2K and Delta q = 17.5% are estimates of the mean errors.

The magnitude of the errors is too high. This is because we are using the partial derivatives incorrectly as described above. Using the partial derivatives described above, the parameter errors are:

Tau_liq: 0.42  tau_ice: 0.28  r_liq: 3.30 um r_ice: 13.08 um  lwp: 2.75 gm-2  iwp: 5.64 gm-2

We reformulated the entire section, improved the description and divided it into several subsections.

**Section 5.7 was confusing at first becase I think the explanation at line 286 is wrong. Table (5) is not "standard deviations of r_ice", but rather the standard deviation of the differences in the retrieved r_ice between two variations of the retrieval that assumed different ice crystal habits. This section would benefit from improved explanation. It is still unclear to me what these results imply about the retrieval product.**

We have removed the entire section. The retrieval was performed again using each ice crystal shape for all spectra. Since neither r_ice of TCWret and Cloudnet correlate independently of ice crystal shape, we see no added value in this section for the interpretation of the results.

**Section 6.4, Line 335: The ice crystal habit selection needs more explanation. If the habit was randomly chosen for r > 30, wouldn't that imply all the habits except droxtal should have a roughly equal percentage of the total retrievals? Was there some other criteria used for selecting the habit (which does not appear in the manuscript?) Also, if the habit is changing between retrievals, then how is this captured in the output product? I do not see any way this was tracked in the output netCDF file.**

In the data, the ice shape is stored as a number. We will add a table with the corresponding meanings in the appendix..

Since the shapes of the ice crystals are not known, the retrievals were carried out for all ice crystal shapes. However, this procedure leads to up to 8 results per measurement, so a selection was made. The accepted result was determined as follows:
- If r_ice for plates, bullet rosettes and solid columns or for droxtals are less than 30 um the result using ice crystals as droxtals is accepted.
- If the first condition is not fulfilled and r_ice for Droxtals are greater than 10 um, the result that uses plates, bullet rosettes or solid columns is accepted. To choose one of the datasets, a random number is drawn which selects plates in 35%, bullet rosettes in 15% and solid columns in 50%
- If none of the conditions above apply, the data for which the degrees of freedom of the outcome are highest is accepted.

The first condition ensures that all small ice particles are classified as droxtals, while the second ensures that all larger particles are classified as plates, solid columns or bullet rosettes. Stricter thresholds would more often result in only the last condition applying, which should be avoided as much as possible.
The ice crystal shapes shown in Figure 9 are determined from this procedure.

**Section 6.3 introduces a cutoff value in PWV (1 cm) which is used to categorize the data. This is based on Cox 2016 ESDD, but the Cox et al manuscript does not address this issue at all. And more importantly, Cox 2016 does not address the water vapor transmission relevant to the specific spectral ranges used in TCWret. A plot or table should be added with the total atmosphere transmission through the selected microwindows at the cutoff value of PWV (1 cm), and I would even add the limit values observed during the campaign (by eye, in Figure 8, this is roughly 0.7 - 1.65 cm).**

Cox et al. (2016, ESSD) is the paper that describes the testcases. The correct paper is Cox et al. (2015, Nature Communications.) We will correct the citation. The limit values are 0.67 cm and 1.62 cm. Since we only mention this cut-off value with reference to the literature, but then also interpret the data with PWV > 1cm, we don't think that a plot the transmission is necessary and rather refer to figure 3 of Cox et al. (2015), where the radiance for PWV = 0.36cm and PWV = 1cm are shown.

**Line 30: the authors quote an LWP uncertainty from a microwave retrieval in the literature; is this using the data from microwave radiometers at the same frequency as Cloudnet? I do not think the MWR frequency for the Cloudnet/OCEANET instruments is mentioned anywhere.**

The LWP of the Cloudnet retrieval is from a HATPRO microwave radiometer and uses two frequency bands (22.24 GHz to 31.4 GHz and 51.0 GHz to 58.0 GHz). The retrieval corresponds to that from Lohnert/Crewell of 2003, as mentioned in Griesche et al.(2020, AMT). With the used frequencies, the RMSE found in Lohnert/Crewell is 22.4 gm-2. We have added the frequencies and the RMSE.

**Line 60: I would suggest adding a couple more simple pieces of information to help understand the dataset: how many days of data were in each "cruise leg", and what was the approximate fraction of time the vessel was in cloudy conditions?**

We will add such information in the section „Observations" based on the number of successful retrievals.

**Minor technical errors, typos, short clarifications, etc:**

**Line 8: "a uncertainty" -> "an uncertainty"**
Done

**Line 12: this is unclear: " ... dataset ... allows to perform ..."**
**suggest " ... dataset ... allows researchers to perform calculations ...", is that the intended meaning?<**
That is true. We rephrased the sentence

**Line 24: "places" -> "place"**
Done

**Line 44: "where low absorption of gases occur" - this is false since the spectral range includes the CO2 absorption band; add a sentence here about the fact that the TCWret is using selected microwindows within that range.**
A sentence that mentions the microwindows is added

**Line 67: "The spectrometer was permanently rinsed with dry air." I have never heard the term "permanently rinsed" used in this context, so this is unclear. Can you explain this in more detail? Is the internal air continuously recirculated with desiccated ambient air, or was it purged with dry air and then sealed during the measurement campaign?**
Continuously dry air was brought into the spectrometer to avoid damage to the hygroscopic beam splitter. We have reformulated it.

**Line 85: what is the length of time for one complete calibration cycle? (specifically, how much time elapses between views of the blackbody at the same temperature?) And what is the duty cycle? (specifically, what fraction of the time is spent looking at the blackbodies versus the atmosphere)**
Each calibration measurement cycle took 10 minutes and each atmosphere measurement took cycle 15 minutes.

**Line 101: "Informations about the cloud ceiling were recorded..." -> "Information about the cloud ceiling was recorded..."**
Done

**Line 124: was the CO2 concentration also the standard atmosphere value, or did you pick a more appropriate value for 2017?**
We used the value for the standard atmosphere (330ppm). This potential source of error is discussed in a section in the appendix. We performed retrievals using spectra from the 11th June 2017 with $CO_2$ concentrations of 330 ppm and 410 ppm. Mean biases and root-mean-square errors are
Tau_liq: -0.0 +- 0.1
Tau_ice: 0.0 +- 0.1
R_liq: 0.1 +- 1.1
R_ice: 0.0 +- 1.3

**Line 133: "Temperature depended" -> "Temperature dependent"**
Done

**Line 138: Were the droplet size and ice crystal size distributions both gamma functions?**
Correct, both size distributions are gamma functions

**Line 159: standard notation for this variable uses "chi", not "xi", and "xi" was already used for the cost function, which is an entirely different quantity. (chi = χ , xi = ξ )**
We changed it from xi to chi

**The expression in (7) is incorrect, assuming this is supposed to be a standard reduced chi^2 variable, it should be:**
chi^2 = Sum( (y - F(x))^2 / sigma^2 ) / DOF
We corrected the expression.

**Line 173: is the retrieval done in log-space, or linear space for tau? (this line seems to contradict what is said just above).**
The retrieval for tau is performed in linear space.

**Line 175: in standard notation, the extinction coefficient is beta, and the extinction cross section is sigma.**
We changed the variables.

**Line 188: By my reading of the Ceccherini and Ridolfi 2010 notation, the left term in parentheses in equation (4) is M_i inverse, not M_i. Please double check.**
That is correct. We have corrected our equation in the manuscript.

**Line 206: Suggest changing the section title to 'Retrieval peformance on simulated data' a similar phrase, to make it clear this section is not using real measurements.**
Done. The new title is as Retrieval performance on simulated spectra

**Line 207: "artifical" -> "artificial"**
Done

**Line 215: "parametern" -> "parameters", "stndard" -> "standard"**
Done

**Table 3: Can you quote the number of test cases used? Is ERR(OE) is the mean of the posterior uncertainty predicted by the OE algorithm?**
ERR(OE) is the mean uncertainty from the covariance matrix. There are 253 test cases included in this analysis. We will mention the number in the caption.

**Line 225: Here, the text states: f_ice = tau_ice * tau_cw, I think this should be f_ice = tau_ice / tau_cw.**
That is true. We have corrected the equation.

**Line 241: This sentence is unclear, could it just be deleted? I am not sure what the authors intend here.**
The sentence is removed.

**Line 249: More detailed is needed. Does this sentence imply that all TCWret retrievals (in particular, those performed on the real measurements from Polarstern) have scaling applied to the posterior errors as predicted by the OE?**
Ja genau, das habe ich mit den Fehlern gemacht. Habe es umformuliert.

**Line 250: "Erorrs" -> "Errors"**
Done

**Line 251: "humidty" -> "humidity"**
Done

**Line 278: equations in text are missing closing parentheses.**
Done

**Line 280: T_BB should be 100 C, not 100 K**
Done

**Figure 5 caption: "retreived" -> "retrieved". Also, these histograms are not the counts, some normalization was done - are these PDFs (meaning they integrate to 1)?**
We have corrected the typo. Also the figure was exchanged. Now it represents the counts.

**Line 330 "intransparent" is not a word. I think what the authors intended to say is "Atmopsheric transmission in the far-infrared spectral region drops to zero for PWV > 1 cm." See earlier comment about this statement.**
That is what this sentence was meant to say. We have exchanged the term to opaque.

**Line 353 "Withouth" -> "Without"**
Done

**Figure 12: The units on the axes are wrong, I think this should be (um)?**
That is correct. The units in the figure have been corrected.

**Line 363 - 365: These sentences are unclear.**
We have rephrased this sentence in the manuscript.

**Line 367: The "very thin clouds" should be the LWP cutoff, not the PWV cutoff?**
That is correct. We have corrected the wording.

**Line 385: "Jupyer" -> "Jupyter"**
Done

**Line 400: I would reiterate that the utilized test cases are simulated or synthetic data, not real observations with some independent estimate.**
Done